



# Comprehensive isoprene and terpene chemistry improves simulated surface ozone in the southeastern U.S.

Rebecca H. Schwantes[1], Louisa K. Emmons[1], John J. Orlando[1], Mary C. Barth[1], Geoffrey S. Tyndall[1], Samuel R. Hall[1], Kirk Ullmann[1], Jason M. St. Clair[2,3], Donald R. Blake[4], Armin Wisthaler[5,6], and ThaoPaul V. Bui[7]

[1]Atmospheric Chemistry Observations and Modeling Laboratory, National Center for Atmospheric Research, Boulder, CO 80301, U.S.A.
[2]Atmospheric Chemistry and Dynamics Laboratory, NASA Goddard Space Flight Center, Greenbelt, MD, 20771, USA
[3]Joint Center for Earth Systems Technology, University of Maryland Baltimore County, Baltimore, MD, 21228, USA
[4]Department of Chemistry, University of California-Irvine, 570 Rowland Hall, Irvine, CA 92697-2025, USA
[5]Institute for Ion Physics and Applied Physics, University of Innsbruck, Technikerstrasse 25, 6020 Innsbruck, Austria
[6]Department of Chemistry, University of Oslo, P.O. 1033 - Blindern, 0315 Oslo, Norway
[7]Earth Science Division, NASA Ames Research Center, Moffett Field, CA 94035-1000

**Abstract.** Ozone is a greenhouse gas and air pollutant that is harmful to human health and plants. During the summer in the southeastern U.S., many regional and global models are biased high for surface ozone compared to observations. Past studies have suggested different solutions including the need for updates to model representation of clouds, chemistry, ozone deposition, and emissions of nitrogen oxides ($NO_x$) or biogenic hydrocarbons. Here due to the high biogenic emissions in

the southeastern U.S., more comprehensive and updated isoprene and terpene chemistry is added into CESM[TM]/CAM-chem (Community Earth System Model/Community Atmosphere Model with chemistry) to evaluate the impact of chemistry on simulated ozone. Comparisons of the model results with data collected during the Studies of Emissions Atmospheric Composition, Clouds and Climate Coupling by Regional Surveys (SEAC[4]RS) field campaign and U.S. EPA CASTNET monitoring stations confirm the updated chemistry improves simulated surface ozone, ozone precursors, and $NO_x$ reservoir compounds. The iso-

prene and terpene chemistry updates reduce the bias in the daily maximum 8-hr average (MDA8) surface ozone by up to 7 ppb. In the past, terpene oxidation in particular has been ignored or heavily reduced in chemical schemes used in many regional and global models, and this study demonstrates comprehensive isoprene and terpene chemistry is needed to reduce surface ozone model biases. Sensitivity tests were performed in order to evaluate the impact of lingering uncertainties in isoprene and terpene oxidation on ozone. Results suggest that even though isoprene emissions are higher than terpene emissions in the southeast-

ern U.S., remaining uncertainties in isoprene and terpene oxidation have similar impacts on ozone due to lower uncertainties in isoprene oxidation. Additionally, this study identifies the need for further constraints on aerosol uptake of organic nitrates derived from isoprene and terpenes in order to reduce uncertainty in simulated ozone.



## 1 Introduction

Many regions of the world have poor air quality due to high levels of tropospheric ozone ($O_3$). Tropospheric ozone is also a greenhouse gas and is an important source of OH radicals, which impacts the lifetime of other greenhouse gases such as methane (Monks et al., 2015; IPCC, 2013). Recent health studies have suggested ozone negatively impacts human health more than previously thought by increasing the risk of both respiratory and circulatory mortality (Turner et al., 2016). Additionally, to protect human health and vegetation in 2015, the U.S. EPA strengthened the ozone standard to not exceed a maximum daily 8 hour average (MDA8) of 70 ppb for more than 3 days a year (U.S.EPA, 2015). Models must accurately simulate ozone for the right reasons to be most effective for predicting future air quality trends (e.g., Val Martin et al., 2015) or to attribute sources of ozone correctly (e.g., Cooper et al., 2015). Because ozone is not directly emitted into the atmosphere and is controlled by large nonlinear sources and losses, ozone is intrinsically difficult to simulate in climate and chemistry models.

Generally, global models capture ozone spatial patterns throughout the troposphere reasonably well, but simulated ozone is typically biased high in the northern hemisphere and low in the southern hemisphere and there are regional and seasonal biases that are not fully understood (Young et al., 2018). During the summer in the southeastern U.S., there is a persistent high bias for surface ozone in many models compared to observations (Fiore et al., 2009; Reidmiller et al., 2009; Brown-Steiner et al., 2015; Tilmes et al., 2015; Canty et al., 2015; Im et al., 2015). Model bias in surface ozone is a good indicator that nitrogen oxides ($NO_x$) or volatile organic compounds (VOCs) budgets and processing are poorly constrained. Past studies have suggested different solutions including the need for updates to model representation of clouds (Ryu et al., 2018), emissions of NO (Travis et al., 2016; McDonald et al., 2018b) or biogenic hydrocarbons (Kaiser et al., 2018), chemistry (Squire et al., 2015), and deposition (Val Martin et al., 2014; Clifton et al., 2019).

Tropospheric ozone is produced in the atmosphere when ozone precursors, anthropogenic or biogenic VOCs and $NO_x$, interact in the presence of sunlight (Monks et al., 2015). The hydroxyl radical (OH) reacts with a VOC to form a peroxy radical ($RO_2$), which reacts with NO to form an organic nitrate or an alkoxy radical and $NO_2$. $NO_2$ will photolyze to form NO and ozone. Organic nitrates are an example of a $NO_x$ reservoir species, a species that has the potential to recycle $NO_x$ back into the system, transport $NO_x$ to a different location, or to permanently remove $NO_x$ from the atmosphere. Correctly representing the production and loss pathways of $NO_x$ reservoir species is critical for accurately representing ozone for the right reasons.

In the southeastern U.S., there are particularly large emissions of biogenic hydrocarbons like isoprene and terpenes, which motivates updating the formation and fate of isoprene and terpene derived organic nitrates in order to assess if more complex and current chemistry reduces model biases in surface ozone. The recent significant improvements in our understanding of isoprene oxidation chemistry (Wennberg et al., 2018, and references therein) have motivated many models to update their isoprene chemistry including GEOS-Chem (Fisher et al., 2016; Bates and Jacob, 2019), GFDL AM3 (Li et al., 2018), MAGRITTEv1.0 (Muller et al., 2018), WRF-Chem (Zare et al., 2018), and CESM2 (Emmons et al., 2019).

Most studies so far have focused on updating isoprene oxidation with significantly less attention to terpenes. In general terpene oxidation has been ignored or heavily reduced in chemical schemes in many regional and global models used in the past despite recent field campaigns suggesting the importance of terpene chemistry (Xu et al., 2015; Zhang et al., 2018). Many





mechanisms represent monoterpenes as a single tracer (e.g., Emmons et al., 2019; Li et al., 2018; Muller et al., 2018). Monoterpenes were expanded to include two surrogate compounds first in WRF-Chem (Browne et al., 2014; Zare et al., 2018) and then in GEOS-Chem (Fisher et al., 2016). These models with expanded terpene chemistry demonstrate the importance of terpene derived organic nitrates for the $NO_x$ budget in both the southeastern U.S. (Fisher et al., 2016) and over the boreal forests of

Canada (Browne et al., 2014). These past studies motivate adding increased complexity for terpenes. Two surrogate compounds are not sufficient to accurately represent all terpene chemistry given the large variety of chemical structures and reactivities (Guenther et al., 2012). Our understanding of terpene chemistry is more limited than isoprene chemistry, but experimental and theoretical data are still available to generate a chemical scheme (Atkinson and Arey, 2003; Johnson and Marston, 2008; Ng et al., 2017, and references therein).

Here, isoprene and terpene chemistry in MOZART-TS1, the default chemical mechanism used in the Community Earth System Model/ Community Atmosphere model with full chemistry (CESM$^{TM}$/CAM-chem), will be updated in order to determine how much chemistry can explain the simulated surface ozone bias over the southeastern U.S. A bias in simulated surface ozone over North America in summer compared to observations was present in past releases of CESM/CAM-chem (Tilmes et al., 2015; Brown-Steiner et al., 2015) and continues to exist in the current release (CESM2.1.0) used in this work (see Section 4.3).

For isoprene, the chemical mechanism updates are of similar complexity to Muller et al. (2018) and Bates and Jacob (2019) and more complex than Travis et al. (2016) and Li et al. (2018). For terpenes, the chemistry updates are significantly more complex than any other reduced scheme currently available (Browne et al., 2014; Fisher et al., 2016; Zare et al., 2018).

The updated isoprene and terpene chemistry will be evaluated against more explicit chemical mechanisms using a box model and against observations using CESM2/CAM-chem. In particular, the formation and fate of the organic nitrates between

the new and old schemes will be described and evaluated. There are a number of lingering uncertainties for both isoprene and terpene chemistry related to the formation and fate of organic nitrates (e.g., differences in measured organic nitrate yields between studies or disagreement among researchers on how to estimate organic nitrate yields for unstudied compounds). These uncertainties will be assessed to determine which uncertainties have the largest impact on simulated surface ozone.

## 2 Development of MOZART-TS2

In this study a new version ("T2") of the MOZART (Model of OZone And Related chemical Tracers) tropospheric chemical mechanism has been developed for use in CESM/CAM-chem and other models. In CAM-chem the T2 mechanism is combined with the current stratospheric mechanism in CESM2 (Emmons et al., 2019), with the result called MOZART-TS2 or "TS2" hereafter. The TS2 mechanism includes a more complex representation of isoprene and terpene oxidation based on recent experimental data than in MOZART-TS1 (Emmons et al., 2019; Knote et al., 2014) or "TS1" hereafter. The updates for isoprene

chemistry include an additional 21 transported species, 18 non-transported species, and 139 reactions, which increases the simulation time by ∼18%. The updates for terpene chemistry include an additional 25 transported species, 22 non-transported species, and 219 reactions, which increases the simulation time by ∼26%. Thus, together these isoprene and terpene updates





increase the simulation time by ∼50%. As described in Section 4.1 and 4.2, this additional cost is necessary in order to correctly simulate $HO_x$ and $NO_x$ recycling and $O_3$ production.

   A list of all TS2 species, photolysis reactions, and kinetic reactions is provided in the Supplement (Tables S2, S5, and S6). A simplified version of TS2 for isoprene and terpene OH-initiated oxidation is shown in Figures 1 and 2 and for $NO_3$-initiated

oxidation in Figures S1 and S2. These figures do not contain all of the detail in TS2, but illustrate the complexity to facilitate comparisons with other reduced schemes and define many of the surrogate species used throughout the text. Explicit chemical mechanisms including MCMv3.3.1 (Jenkin et al., 2015) and the Caltech isoprene mechanism (Wennberg et al., 2018) and several review papers (Atkinson and Arey, 2003; Johnson and Marston, 2008; Ng et al., 2017) strongly guided the creation of the reduced TS2 mechanism. Surrogate compounds are shared from $NO_3$, $O_3$, and OH-initiated oxidation to ensure accurate

representation of the chemistry while reducing the number of surrogate compounds and the computational cost.

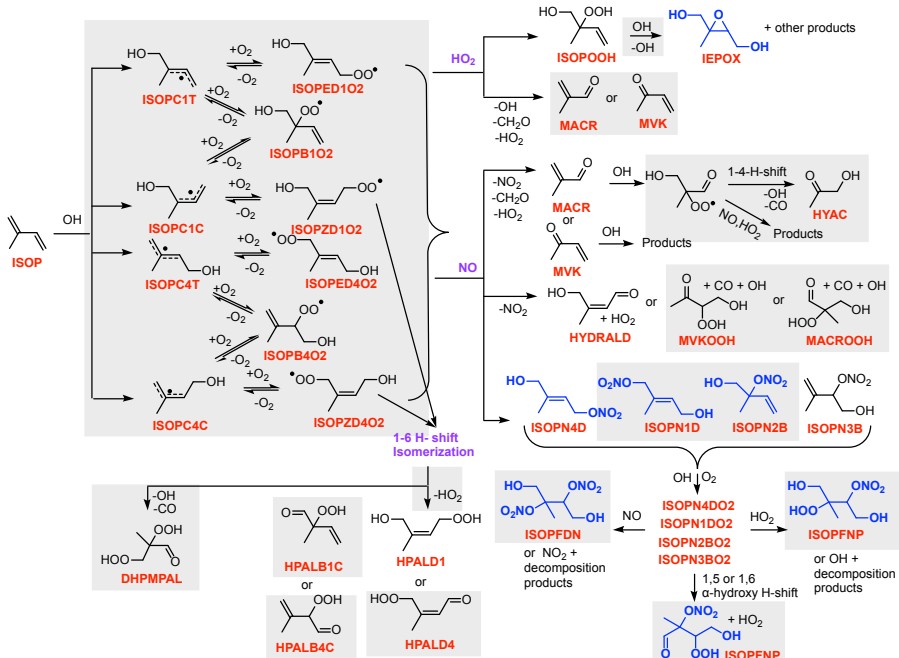

**Figure 1.** Simplified schematic of the TS2 mechanism for isoprene OH-initiated oxidation. Gray boxes indicates new chemistry added or updated in TS2. Blue compounds undergo aerosol uptake.

   When available, all reaction rate constants were updated to those recommended in either JPL (Burkholder et al., 2015) or IUPAC (Atkinson et al., 2004, 2006). For those reaction rates not in IUPAC or JPL, typically the Caltech isoprene mechanism (Wennberg et al., 2018) or MCM v3.3.1 (Jenkin et al., 1997; Saunders et al., 2003; Jenkin et al., 2012, 2015) was used. For isoprene and terpene reactions, the peroxy ($RO_2$) and peroxyacyl ($RCO_3$) reaction rates were consistently assigned throughout

the mechanism using the assumptions specified in Table S1.





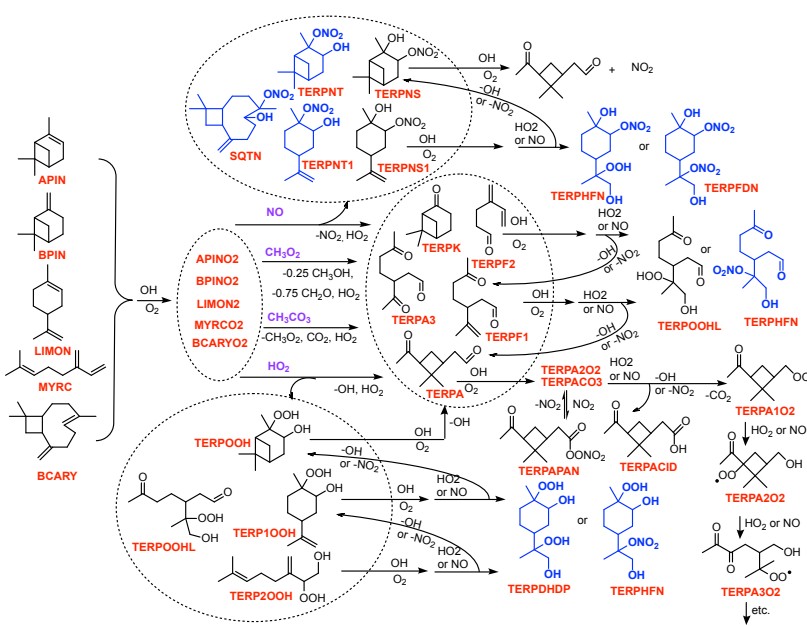

**Figure 2.** Simplified schematic of the TS2 chemical mechanism for terpene OH-initiated oxidation. Blue compounds undergo aerosol uptake.

## 2.1 Updates to Henry's Law Constants

Currently, in TS1, only certain species undergo wet and dry deposition (Emmons et al., 2019). For TS2, all compounds undergo wet and dry deposition except for radicals and compounds constrained with lower-boundary conditions. As listed in Table S4, Henry's law constants were updated to the most recent literature recommendations (Burkholder et al., 2015; Sander, 2015;

Schwartz and White, 1981; Leu and Zhang, 1999; Goldstein and Czapski, 1997; Fried et al., 1994; Chameides, 1984; Reichl, 1995; Kames and Schurath, 1995; Leng et al., 2013; Chan et al., 2010; Staudinger and Roberts, 2001; Dohnal and Fenclova, 1995; Hiatt, 2013; Guo and Brimblecombe, 2007; McNeill et al., 2012; Allou et al., 2011; Sieg et al., 2009; Iraci et al., 1999; Smith and Martell, 1976; Copolovici and Niinemets, 2005; van Roon et al., 2005). The effective Henry's law calculations used in CAM-chem are described in the notes at the end of Table S4. Henry's law constants for halogens important mainly for

stratospheric chemistry were not changed from previous versions (Emmons et al., 2019). For all oxygenated organic gases that condense to form SOA, Henry's law coefficients were based on values from GECKO-A as in Hodzic et al. (2014, 2016) with no changes from previous versions (Emmons et al., 2019). When Henry's law constants were unavailable in the literature, the value was approximated based on a close surrogate or by GROMHE (Raventos-Duran et al., 2010). GROMHE is the theoretical structure activity relationship method used to estimate Henry's law constants by the Generator of Explicit Chemistry and

Kinetics for Organics in the Atmosphere (GECKO-A) (Aumont et al., 2005). If the Henry's law temperature dependence was unavailable in the literature, 6014 K was assumed consistent with GECKO-A. The reactivity factor ($F_0$), which ranges from 0





to 1 with 1 being as reactive as ozone, is also listed in Table S4. The $F_0$ for oxygenated volatile organic compounds is assumed to be 1 consistent with recent observational studies (Karl et al., 2010; Nguyen et al., 2015).

## 2.2 Updates to Isoprene Chemistry

Isoprene oxidation by OH (Section 2.2.1), $O_3$ (Section 2.2.2), and $NO_3$ (Section 2.2.3) were all updated in TS2 from TS1.
All new photolysis reactions were mapped with an optional scaling factor to photolysis rate constants already incorporated into CESM2. Scaling to known photolysis rates is common in reduced chemical mechanisms and even explicit mechanisms like MCM as photolysis rates for many surrogate compounds have not been measured. In general, products and photolysis rate constants were guided by explicit schemes: MCM v3.3.1 (Jenkin et al., 2015) and the Caltech isoprene mechanism (Wennberg et al., 2018). $\delta$-hydroperoxy aldehydes (HPALD1 and HPALD2) were assumed to photolyze with the cross sections of methacrolein (Wennberg et al., 2018) and the quantum yield estimated by Liu et al. (2017). Carbonyl nitrates were assumed to photolyze with the fast photolysis rate constants reported in Muller et al. (2014). Like MCM v3.3.1 (Jenkin et al., 2015), the various carbonyl nitrate photolysis rates are scaled to that of propanone nitrate (NOA). The photolysis rate constant for isoprene carbonyl nitrate from isoprene $NO_3$-initiated oxidation (NC4CHO) is based on the measurement from Xiong et al. (2016).

### 2.2.1 OH-Initiated Oxidation

Isoprene reacts with OH and then $O_2$ to form 6 distinct isoprene hydroxy peroxy radicals (Figure 1), which are represented explicitly in TS2 based on reaction rate constants reported by Teng et al. (2017). The Caltech isoprene mechanism recommends a possible reduction to represent this 1$^{st}$-generation peroxy radical chemistry, but this reduction scheme does not perform as well in urban regions with high NO and short $RO_2$ lifetimes (Wennberg et al., 2018). Here this chemistry is represented explicitly (i.e. 4 isoprene hydroxy alkyl radical isomers and 6 isoprene hydroxy peroxy radical isomers) because radical species are not transported in CESM/CAM-chem and so do not considerably contribute to the computational cost. This more explicit chemistry allows TS2 to be used at finer horizontal resolutions that better resolve urban regions with high NO levels, which is a goal for future studies. The $\delta$-Z-isoprene hydroxy peroxy radicals will isomerize in TS2 to form four isomers (2-$\beta$ and 2-$\delta$) of hydroperoxy aldehydes (HPALDs) among other products based on recommendations from Wennberg et al. (2018), but more reduced. There are still large uncertainties in the rates, product yields, and $HO_x$ recycling from photolysis and OH oxidation of HPALDs. Once more is known, greater detail may be added to TS2.

While TS1 assumes a unity yield of isoprene hydroxy hydroperoxide (ISOPOOH) from the isoprene $RO_2$ + $HO_2$ reaction, TS2 adds a small yield of methyl vinyl ketone and methacrolein from this pathway (Liu et al., 2013; Wennberg et al., 2018). The ISOPOOH + OH reaction rate and products have been updated to be consistent with St. Clair et al. (2016). Chemistry for isoprene epoxydiol (IEPOX), the dominant product from ISOPOOH + OH reaction, has also been updated in TS2 (Bates et al., 2014, 2016; Jacobs et al., 2013; Wennberg et al., 2018). TS2 only has one isomer of ISOPOOH and IEPOX, which is more reduced than the Caltech mechanism (Wennberg et al., 2018), but this simplification has minimal impact on the total



ISOPOOH or IEPOX concentration (Section 4.1). To save computational cost, multiple isomers were only incorporated into TS2 if grouping them together would bias the $HO_x$ or $NO_x$ budgets (e.g., HPALDs and hydroxy nitrates).

In TS2, additional organic nitrates are added to better represent $NO_x$ recycling in the mechanism. Four (ISOPN4D, ISOPN1D, ISOPN2B, ISOPN3B) instead of two 1st-generation hydroxy nitrate isomers are included using the temperature and pressure dependent yields recommended by Wennberg et al. (2018) and Teng et al. (2017) ($\sim$0.13 at 297K). Often different isomers of the same compound will have very different fates in the atmosphere. For example, $\beta$ and $\delta$-hydroxy nitrates have different reaction rates with OH and $O_3$ and form different products (Lee et al., 2014). Additionally, tertiary organic nitrates will be more likely to undergo aerosol uptake due to their rapid hydrolysis in the particle phase compared to secondary or primary organic nitrates (Section 2.4). The number of organic nitrates in TS2 was optimized to accurately represent these varying fates. The differences in organic nitrate formation and fate between TS1 and TS2 are described in more detail in Section 4.5.

In TS1, OH oxidation of the 1st-generation hydroxy nitrates immediately forms stable products from the $RO_2$ + NO channel rather than going through a peroxy radical intermediate. Conversely, in TS2, when an isoprene hydroxy nitrate is oxidized by OH, a peroxy radical forms, which can either isomerize or react with NO or $HO_2$ (Jacobs et al., 2014; Lee et al., 2014; Wennberg et al., 2018). Because $NO_x$ emissions are generally decreasing or expected to decrease in the United States (Kharol et al., 2015; EPA, 2018; Jiang et al., 2018) and mixed regimes are becoming more prevalent, chemical mechanisms that do not fix the fate of the peroxy radical to the 1st-generation fate will become increasingly important. For example, in TS2, peroxy radicals from OH oxidation of unsaturated organic nitrates produced from the $RO_2$ + NO channel and IEPOX produced from the $RO_2$ + $HO_2$ channel can react either with NO or $HO_2$ in the later-generation step. A wide variety of later-generation organic nitrates are added to TS2 including those from decomposition ($C_2$-$C_4$) and functionalization (ISOPFNP and ISOPFDN in Figure 1). Including additional surrogate compounds for highly functionalized nitrates, whose fates are likely to remove $NO_x$ through aerosol uptake/hydrolysis or wet/dry deposition, is important for accurately representing the $NO_x$ budget.

Methacrolein and methyl vinyl ketone, which are 1st-generation products from isoprene, react with OH to form separate peroxy radicals in TS2 (Figure 1), so that the methacrolein hydroxy peroxy radical can undergo an isomerization reaction (Crounse et al., 2012). In TS2, methacrolein and methyl vinyl ketone peroxy radicals react with $HO_2$ to form not only hydroxy hydroperoxides (MACROOH and MVKOOH) like in TS1, but also OH and decomposition products (Praske et al., 2015; Wennberg et al., 2018). The organic nitrate yields from the methacrolein and methyl vinyl ketone $RO_2$ + NO reactions have been updated and form distinct organic nitrates (MACRN and MVKN) (Crounse et al., 2012; Praske et al., 2015). Products from the reaction of $HO_2$ with all acyl peroxy radicals including the one derived from methacrolein were updated to IUPAC recommendations (Atkinson et al., 2006) based on the following studies (Hasson et al., 2004; Jenkin et al., 2007; Dillon and Crowley, 2008; Niki et al., 1985; Horie and Moortgat, 1992) and averaging in results from a more recent study (Grob et al., 2014).

### 2.2.2 $O_3$-Initiated Oxidation

$O_3$-initiated oxidation of isoprene is largely based on Grosjean et al. (1993a), Aschmann and Atkinson (1994), Nguyen et al. (2016), and IUPAC (Atkinson et al., 2006). Given isoprene is typically emitted in regions with high RH, the stabilized criegees





are not represented explicitly in the mechanism and instead are assumed to react immediately with $H_2O$ and $(H_2O)_2$ in equal amounts, which is approximately the case for RH > 60%, to form hydroxymethyl hydroperoxide (HMHP), formaldehyde and $H_2O_2$, or formic acid (Nguyen et al., 2016). HMHP, a new product in TS2, photolyzes and reacts with OH based on Roehl et al. (2007), Allen et al. (2018), and Wennberg et al. (2018). The reaction of $HOCH_2OO + HO_2$, which includes HMHP as a product, was updated to IUPAC recommendations (Atkinson et al., 2006).

### 2.2.3 $NO_3$-Initiated Oxidation

Isoprene $NO_3$-initiated oxidation is largely based on the following studies: Schwantes et al. (2015) and Wennberg et al. (2018) for 1st-generation and Schwantes et al. (2015), Lee et al. (2014), Xiong et al. (2016), and Jacobs et al. (2014) for 2nd-generation products. Both TS1 and TS2 only have one surrogate compound for the nitrooxy peroxy radical formed when isoprene reacts with $NO_3$. Based on the products formed, TS1 assumes all the nitrooxy peroxy radicals are $\delta$-isomers while TS2 uses the yields in Schwantes et al. (2015), which estimated approximately equal amounts of $\beta$- and $\delta$-nitrooxy peroxy radicals. Additionally, TS2 assumes a non-unity yield of nitrooxy hydroperoxide from the $RO_2 + HO_2$ channel consistent with recent work (Schwantes et al., 2015; Wennberg et al., 2018; Rollins et al., 2009; Kwan et al., 2012). Where structurally similar, organic nitrates from the OH and $NO_3$ system are shared to reduce computational cost. Two isomers ($\beta$ and a $\delta$) of isoprene nitrooxy hydroxy epoxide formed from OH oxidation of nitrooxy hydroperoxide are added to TS2 (Schwantes et al., 2015; Wennberg et al., 2018). Similar to OH-initiated oxidation, the organic nitrates derived from $NO_3$-initiated oxidation react with OH to form a peroxy radical that can isomerize or react with NO or $HO_2$ (Lee et al., 2014; Jacobs et al., 2014; Schwantes et al., 2015; Wennberg et al., 2018; Xiong et al., 2016).

### 2.3 Updates to Terpene Chemistry

In TS1, all monoterpenes are grouped into one surrogate (MTERP) and all sesquiterpenes are grouped into one surrogate (BCARY). The OH, $O_3$, and $NO_3$ reaction rate constants are different between MTERP and BCARY, but the oxidation chemistry is identical. An expanded version of the TS1 terpene chemistry used primarily in WRF-Chem, called T1 (Emmons et al., 2019; Knote et al., 2015a), replaced MTERP with four monoterpene surrogates: $\alpha$-pinene (APIN), $\beta$-pinene (BPIN), limonene (LIMON) and myrcene (MYRC). In T1, the five terpene surrogates have different reaction rates with OH, $O_3$, and $NO_3$, but their oxidation chemistry is identical. Here in TS2, we start from the T1 mechanism and group all rather than a subset of the terpenes in MEGAN v2.1 (online in CESM) according to their chemical structure and reactivity into the 5 terpene surrogate compounds: APIN, BPIN, LIMON, MYRC, and BCARY (Table S3). Unlike T1, in TS2 each terpene surrogate has unique chemistry. Even though the chemistry is different for each terpene surrogate in TS2, the 1st- and later-generation products are often shared to save computational cost. For example, the terpene hydroxy nitrate surrogate compounds are split according to their chemical structure (saturated versus unsaturated and primary/secondary versus tertiary) instead of based on their VOC precursor (i.e. there are not unique APIN hydroxy nitrates and BPIN hydroxy nitrates). Terpene oxidation chemistry by OH (Section 2.3.1), $O_3$ (Section 2.3.2), and $NO_3$ (Section 2.3.3) were all updated in TS2 from TS1. Like isoprene, rates for all pho-




tolysis reactions were mapped to photolysis rate constants already incorporated into CESM2 (Table S5). In general, photolysis reactions and rate constants were guided by MCM v3.3.1 (Jenkin et al., 2015).

### 2.3.1 OH-Initiated Oxidation

The terpene surrogate compounds react with OH to form hydroxy peroxy radicals that react with NO, $NO_3$, $RO_2$, or $HO_2$. For

APIN, BPIN, LIMON, MYRC, and BCARY, the products from $RO_2$ + NO reaction, which have been reasonably well studied, were used to extrapolate the products from the $RO_2$ + $NO_3$ and $RO_2$ + $RO_2$ reactions. The hydroxy hydroperoxide yield from the $RO_2$ + $HO_2$ channel has only been measured for $\alpha$-pinene (Noziere et al., 1999; Eddingsaas et al., 2012). For the rest, the hydroxy hydroperoxide yield is estimated based on the parameterization recommended by Wennberg et al. (2018) assuming the same $RO_2$ distribution used in the $RO_2$ + NO reaction. Below we explain briefly the experimental and theoretical studies

used to determine the product distribution from the $RO_2$ + NO reaction for each terpene surrogate compound. Organic nitrate yields from the $RO_2$ + NO pathway for many later-generation terpene oxidation products (e.g., pinonaldehyde, nopinone, and limonaldehyde) have not been measured; in this case, the organic nitrate yield is approximated from the parameterization by Wennberg et al. (2018) up to a maximum of 0.3. Experimental work on the alkane system suggests that the organic nitrate yield plateaus at 0.3 (Arey et al., 2001; Yeh and Ziemann, 2014). No past literature studies have evaluated whether this plateau is

different for oxygenated VOCs, which could have important consequences on ozone. Future experimental studies constraining organic nitrate yields from oxygenated VOCs is highly recommended for further improvement of the TS2 mechanism. Due to the large uncertainties in the organic nitrate yields from terpene oxidation, no temperature or pressure dependency was included.

The APIN $RO_2$ will react with NO to form hydroxy nitrates (yield = 0.23) (Noziere et al., 1999; Ruppert et al., 1999;

Rindelaub et al., 2015), acetone (Aschmann et al., 1998; Noziere et al., 1999; Orlando et al., 2000; Wisthaler et al., 2001), formaldehyde (Noziere et al., 1999; Orlando et al., 2000), pinonaldehyde (TERPA) (Arey et al., 1990; Hakola et al., 1994; Wisthaler et al., 2001; Aschmann et al., 2002a), and the remaining products are based on theoretical work (Vereecken et al., 2007). The hydroxy nitrate isomer distribution is based on one experimental study (Berndt et al., 2016) and theoretical work (Vereecken et al., 2007). The products from the $RO_2$ + $HO_2$ channel (TERPOOH, TERP1OOH, TERPA) are based on experi-

mental (Noziere et al., 1999; Eddingsaas et al., 2012) and theoretical (Vereecken et al., 2007) studies. Fewer studies have been conducted on $\beta$-pinene than $\alpha$-pinene, but still enough information is available to develop a scheme. The BPIN $RO_2$ will react with NO to form nopinone (TERPK) (Arey et al., 1990; Hakola et al., 1994; Wisthaler et al., 2001; Lee et al., 2006b; Jaoui and Kamens, 2003), formaldehyde (Hatakeyama et al., 1991; Orlando et al., 2000; Lee et al., 2006b), acetone (Aschmann et al., 1998; Orlando et al., 2000; Wisthaler et al., 2001; Lee et al., 2006b), hydroxy nitrates (yield = 0.25) (Ruppert et al., 1999),

and the remaining products are based on theory (Vereecken and Peeters, 2012). The $RO_2$ isomer distribution is based on the measured product yields defined above for hydroxy nitrates and nopinone and theoretical work from Vereecken and Peeters (2012).

The LIMON $RO_2$ isomer distribution is approximated using the SAR developed by Peeters et al. (2007). These $RO_2$ will react with NO to form hydroxy nitrates (yield = 0.23) (Ruppert et al., 1999), formaldehyde (Lee et al., 2006b), and a terpene





oxidation product containing one double bond (TERPF1), which represents both limonaldehyde and limaketone. MYRC $RO_2$ isomer distribution is also approximated by the SAR developed by Peeters et al. (2007) and reacts with NO to form hydroxy nitrates (yield = 0.29) (Ruppert et al., 1999), acetone (Reissell et al., 1999; Orlando et al., 2000; Lee et al., 2006b), formaldehyde (Orlando et al., 2000), and the main product is assumed to be TERPF2, which is a functionalized terpene oxidation product with

two double bonds. OH-initiated oxidation is quite uncertain for $\beta$-caryophyllene, the surrogate for all sesquiterpenes. From the BCARY $RO_2$ reaction with NO we form a sesquiterpene nitrate called SQTN (yield = 0.3) and TERPF2. The sesquiterpenes have 2-4 double bonds (Guenther et al., 2012), so we assume TERPF2 forms as the primary 1st-generation product. Because SQTN has at least one double bond and is low in volatility, it or its oxidation products, which will retain the nitrate group, will likely deposit or undergo aerosol uptake. Thus, we do not include further reaction of SQTN with OH or photolysis.

In order to save computational cost and still represent the chemistry as accurately as possible, the 1st- and later-generation products for all the terpene surrogate compounds are shared. There are a number of aldehyde surrogate compounds including: TERPA, which represents pinonaldehyde type products; TERPA2, which represents norpinaldehyde type products; and TERPA3, which represent aldehydes largely produced from limonaldehyde and limaketone oxidation. These aldehyde products react with OH, $O_3$, and $NO_3$ based on pinonaldehyde from MCM v3.3.1 (Saunders et al., 2003). Each of these will

form corresponding carboxylic/peroxy acids (TERPACID,TERPACID2, TERPACID3) and PANs (TERPAPAN, TERPA2PAN, TERPA3PAN). One ketone is included: TERPK, which represents nopinone type products and reacts with OH using the rate from Calogirou et al. (1999). There are two unsaturated compounds, which represent functionalized terpene oxidation products with one (TERPF1) or two (TERPF2) double bonds. TERPF1 reacts with OH, $O_3$, and $NO_3$ like limonaldehyde (Calogirou et al., 1999) and TERPF2 reacts with OH and $NO_3$ like isoprene (Atkinson et al., 2006) and $O_3$ like 1st-generation prod-

ucts from $\beta$-caryophyllene (Winterhalter et al., 2009). There are three hydroperoxides: TERPOOH, which represents saturated hydroxy hydroperoxides with a ring (e.g., from $\alpha$-pinene); TERPOOHL, which represents saturated hydroxy hydroperoxides without a ring (e.g., from limonene); TERP1OOH, which represent hydroxy hydroperoxides with one double bond; and TERP2AOOH, which represent hydroxy hydroperoxides with two double bonds. In order to represent terpenes with multiple double bonds accurately, the later-generation products must continue to contain a double bond that undergoes OH-addition.

There are four 1st-generation hydroxy nitrates: TERPNS (primary and secondary saturated), TERPNS1 (primary and secondary unsaturated), TERPNT (tertiary saturated), and TERPNT1 (tertiary unsaturated). Saturated and unsaturated hydroxy nitrates are separated because they have different reaction rates and products from oxidation by OH. Tertiary hydroxy nitrates are separated from primary/secondary hydroxy nitrates because tertiary nitrates will undergo aerosol uptake due to their rapid hydrolysis (Section 2.4). There are also a number of low-volatility highly functionalized compounds such as TERPFDN, which

represents highly functionalized terpene dinitrates; TERPHFN, which represents highly functionalized terpene nitrates; and TERPDHDP, which represents terpene dihydroxy dihydroperoxides.

### 2.3.2   $O_3$-Initiated Oxidation

$O_3$-initiated oxidation of APIN is based on MCM (Saunders et al., 2003), IUPAC (Atkinson et al., 2006), theoretical calculations (Zhang and Zhang, 2005; Kurten et al., 2015), and experimental results (Ma et al., 2008). The OH yield (0.77) is quite





high (Chew and Atkinson, 1996; Paulson et al., 1998; Rickard et al., 1999; Siese et al., 2001; Berndt et al., 2003; Forester and Wells, 2011). The stabilized criegees are not represented explicitly in the mechanism and instead are assumed to react immediately with $H_2O$ to form either $H_2O_2$ and pinonaldehyde (TERPA) or pinonic acid (TERPACID). $O_3$-initiated oxidation of BPIN is based on MCM (Saunders et al., 2003), IUPAC (Atkinson et al., 2006), experimental results (Hakola et al., 1994;

Grosjean et al., 1993b; Yu et al., 1999; Ma and Marston, 2008; Winterhalter et al., 2000; Hasson et al., 2001), and theory (Nguyen et al., 2009). The OH yield (0.3) from $\beta$-pinene ozonolysis is much lower than that from $\alpha$-pinene due to differences in their molecular structures (Atkinson et al., 1992; Rickard et al., 1999). Again the stabilized criegee intermediates are not explicitly represented and instead assumed to react directly with $H_2O$ to form $H_2O_2$ and nopinone (TERPK).

Only a few products with low yields have been detected from limonene ozonolysis (Atkinson and Arey, 2003). The chemical

mechanism for $O_3$-initiated LIMON oxidation is based on MCM (Saunders et al., 2003) and IUPAC (Atkinson et al., 2006). The OH yield is 0.66 (Aschmann et al., 2002a; Forester and Wells, 2011; Herrmann et al., 2010). The majority of products are grouped into one surrogate species, TERPF1, which is a terpene functionalized oxidation product with one double bond. Again stabilized criegee intermediates were not included explicitly and instead assumed to react with $H_2O$ to form $H_2O_2$ and TERPF1 or TERPACID in similar yields to that in the $\alpha$-pinene system. $O_3$-initiated oxidation of MYRC is based on Ruppert

et al. (1999), Boge et al. (2013) and Lee et al. (2006a) with an OH yield of 0.63 from Aschmann et al. (2002a). The dominant products are hydroxy acetone (HYAC), acetone, and 4-vinyl-4-pentenal (TERPF2). $O_3$-initiated oxidation of BCARY is based on Winterhalter et al. (2009) and Jaoui et al. (2003) with an OH yield of 0.08 (Shu and Atkinson, 1994; Winterhalter et al., 2009) to form TERPACID and TERPF2.

### 2.3.3 NO$_3$-Initiated Oxidation

For NO$_3$-initiated oxidation, $\alpha$-pinene has been studied the most completely. Based on the few studies investigating the other monoterpenes, $\alpha$-pinene oxidation by NO$_3$ is unique (Hallquist et al., 1999; Fry et al., 2014; Kurten et al., 2017), so $\alpha$-pinene is handled separately in TS2 as APIN. In TS2, monoterpenes react with NO$_3$ to form a nitrooxy peroxy radical, which then reacts with HO$_2$, NO, NO$_3$, CH$_3$CO$_3$, CH$_3$O$_2$, and itself to form different yields of organic nitrates. There are few experimental studies that explicitly state the RO$_2$ fate, when reporting a nitrate yield. In all future experiments, reporting

the RO$_2$ fate is recommended, so that laboratory results can be directly used in the development of condensed and explicit chemical mechanisms.

In TS1, when the nitrooxy peroxy radical reacts with HO$_2$, one nitrooxy hydroperoxide isomer forms with unity yield. In contrast, TS2 forms four nitrooxy hydroperoxide isomers: TERPNPT (saturated tertiary), TERPNPS (saturated secondary/primary), TERPNPT1 (unsaturated tertiary), and TERPNPS1 (unsaturated primary/secondary). Recent work suggests that the $\alpha$-pinene

nitrooxy peroxy radical reacts with HO$_2$ to form nitrooxy hydroperoxide (0.3) and pinonaldehyde (0.7) (Kurten et al., 2017). The yield of nitrooxy hydroperoxides for all other surrogate terpene compounds is estimated from the parameterization in Wennberg et al. (2018). Berndt and Boge (1997) measured a 0.14 nitrate or 0.07 dinitrate yield from $\alpha$-pinene nitrooxy peroxy + NO reaction. Because the $\alpha$-pinene nitrooxy alkoxy radical is unlikely to decompose to form organic nitrates (Fry et al., 2014; Kurten et al., 2017), this signal is assumed to be from dinitrates. Thus, in TS2, all terpene surrogate nitrooxy peroxy




radicals react with NO to form a yield of 0.07 dinitrates. Most $NO_3$-initiated oxidation laboratory experiments have focused on $RO_2 + RO_2$ chemistry. The fate of the alkoxy radical can be inferred from these product distributions and used to estimate the oxidation products from the other pathways (i.e., $RO_2 + HO_2$, $RO_2 + NO_3$, and $RO_2 + NO$).

APIN nitrooxy peroxy radical + $RO_2$ reactions form organic nitrates and pinonaldehyde (TERPA) (Wangberg et al., 1997;
Hallquist et al., 1999; Spittler et al., 2006). The nitrooxy peroxy radical isomer distribution is based on MCM v3.3.1 (Saunders et al., 2003). The APIN nitrooxy alkoxy radical is assumed not to form any organic nitrates. For BPIN, the tertiary peroxy radical (yield of 0.9) is dominant (Boyd et al., 2015), so the $RO_2 + RO_2$ reactions are presumed to mostly form alkoxy radicals rather than carbonyl or hydroxy nitrates. Nopinone is produced in a low yield (0.02) (Hallquist et al., 1999), and the remaining products are quite uncertain. The alkoxy radical from $\delta$-3-carene, which is grouped with $\beta$-pinene, breaks preferentially at the
$C-C(H_2)$ bond to retain the nitrate group rather than the $C-C(ONO_2)$ bond to release $NO_2$ (Kurten et al., 2017). For $RO_2 +$ $RO_2$ reactions from BPIN, the organic nitrate yield is based on Hallquist et al. (1999) and Fry et al. (2009). The BPIN alkoxy radical is assumed to decompose to form organic nitrates based on the average between $\beta$-pinene and $\delta$-3-carene, which are grouped into one surrogate compound (Fry et al., 2014).

LIMON $NO_3$-initiated oxidation is more complicated because of the 2 double bonds. Spittler et al. (2006) determined that
$NO_3$ addition is more selective than OH, and so $NO_3$ dominantly reacts with the endocylic double bond of limonene. LIMON $NO_3$-initiated oxidation was assumed to form a similar initial nitrooxy peroxy radical distribution as $\alpha$-pinene. The organic nitrate yield for $RO_2 + RO_2$ reactions is based on Hallquist et al. (1999), Spittler et al. (2006) and Fry et al. (2014). MYRC $NO_3$-initiated oxidation has not been constrained by any studies. MYRC nitrooxy peroxy radical distribution was calculated based on SARs (Pfrang et al., 2006) and the following assumptions. For conjugated double bonds, $NO_3$ adds to the less
substituted position and equal amounts of $\delta$- and $\beta$-peroxy radicals form consistent with isoprene oxidation (Teng et al., 2017; Schwantes et al., 2015). For the non-conjugated double bond, $NO_3$ adds in the same ratio as that for $\alpha$-pinene. For organic nitrate yields, all secondary/tertiary alkoxy radicals were assumed to release $NO_2$ and all primary alkoxy radicals were assumed to produce carbonyl nitrates. The $RO_2 + RO_2$ reaction was assumed to form the same organic nitrate yield as $\alpha$-pinene. All of these assumptions are quite speculative. Better understanding of less studied monoterpenes like myrcene is necessary.

There are also few constraints on $NO_3$-initiated oxidation of $\beta$-caryophyllene. All sesquiterpene derived organic nitrates are grouped together as SQTN consistent with OH-initiated oxidation. Fry et al. (2014) detected all of the organic nitrates from $\beta$-caryophyllene in the particle phase. Thus, the main loss of SQTN in the atmosphere and in TS2 is aerosol uptake and wet/dry deposition to permanently remove $NO_x$ in the atmosphere. The BCARY peroxy radical distribution and organic nitrate yields were assumed to be similar to that of $\alpha$-pinene. Sesquiterpene $NO_3$-initiated oxidation is quite uncertain and difficult to
constrain given current literature data. However, sesquiterpene derived nitrates, which quickly partition to the particle-phase or deposit on surfaces, may be important missing removal pathways for $NO_x$ and so deserve further study.

The four nitrooxy hydroperoxides derived from $NO_3$-initiated oxidation react with OH and photolyze largely based on MCM v3.3.1 (Saunders et al., 2003). However, in MCM v3.3.1, unsaturated nitrooxy hydroperoxides react with OH via hydrogen abstraction rather than addition to the double bond. This over-simplification is common in later-generation oxidation pathways
in MCM v3.3.1, and leads to inaccurate $NO_x$ recycling (Section 4.2). Instead of using MCM v3.3.1 recommendations, OH is



assumed to react with the unsaturated nitrooxy hydroperoxides at the same rate as limonaldehyde (Calogirou et al., 1999) and to largely form the corresponding saturated nitrooxy hydroperoxide.

## 2.4 Aerosol Uptake of Isoprene and Terpene Organic Nitrates

In CESM/CAM-chem, uptake of gas-phase organic nitrates to aerosols is represented simply by converting an organic nitrate to nitric acid thereby neglecting the entire particle-phase hydrolysis reaction. TS1 uses aerosol uptake parameters based on Fisher et al. (2016) for isoprene ($\gamma = 0.005$) and terpene ($\gamma = 0.01$) derived organic nitrates. Fisher et al. (2016) recommends a bulk aerosol uptake coefficient for all isoprene hydroxy nitrate isomers, even though only tertiary and $\delta$-allylic-hydroxy nitrates will undergo fast enough hydrolysis for aerosol uptake to be relevant in the atmosphere (Jacobs et al., 2014). In TS2, only tertiary isoprene and terpene organic nitrates and isoprene $\delta$-allylic-hydroxy nitrates undergo aerosol uptake (Jacobs et al., 2014), but with a larger aerosol uptake coefficient ($\gamma = 0.02$) recommended by Wolfe et al. (2015). The newly added multifunctional isoprene and terpene low-volatility organic nitrates in TS2 undergo rapid aerosol uptake ($\gamma = 0.1$) based on Marais et al. (2016). Aerosol uptake of organic nitrates is quite uncertain (Section 4.6) and deserves further study.

## 3 Methods

The newly developed TS2 mechanism was evaluated against explicit mechanisms using a box model (Section 3.1) and against field observations using CESM/CAM-chem (Section 3.2).

### 3.1 Box Modeling

MOZART-TS2 was compared with MOZART-TS1, MCM v3.3.1 (Jenkin et al., 2015), and the Caltech isoprene mechanism (Wennberg et al., 2018) using BOXMOX v1.7 (Knote et al., 2015b), a box model wrapper for the Kinetic Preprocessor (KPP) (Sandu and Sander, 2006). The Master Chemical Mechanism, MCM v3.3.1, was downloaded via the website: http://mcm.leeds.ac.uk/MCM, last access: 7 September 2018 (Jenkin et al., 1997; Saunders et al., 2003; Jenkin et al., 2012, 2015). The Caltech mechanism (RCIM), isoprene reduced plus v4.1, was downloaded from http://doi.org/10.22002/D1.247, last access: 23 March 2018 (Bates and Wennberg, 2017). The inorganic reactions from MCM v3.3.1 were used for all mechanisms in order to focus on differences caused by isoprene and terpene chemistry. To capture differences in OH, $O_3$, and $NO_3$ oxidation, an ideal diurnal cycle was simulated in the box model with the planetary boundary layer, temperature, photolysis rate constants, emissions of NO, CO, isoprene, formaldehyde, formic acid, methanol, glycolaldehyde, sulfur dioxide, sesquiterpenes, and monoterpenes from the CESM/CAM-chem base TS1 simulation. Aerosol uptake of the following inorganic compounds: $HO_2$, $N_2O_5$, $NO_2$, $NO_3$ and deposition of the following inorganic compounds: $O_3$, CO, NO, $NO_2$, $HNO_3$, $N_2O_5$, $HO_2NO_2$, $H_2O_2$, $SO_2$ were also from the CESM/CAM-chem base simulation.

Each mechanism (Caltech, MCM, and TS1/TS2) calculates photolysis rates differently. In CESM/CAM-chem, general photolysis rates are calculated using a lookup table (Lamarque et al., 2012) and all other photolysis rates are mapped to these general photolysis rates with an optional scaling factor. In the box model setup, general photolysis rates from CESM/CAM-





chem are used for all mechanisms and the scaling factors are mechanism specific. This approach ensures consistency in the general photolysis reactions across mechanisms, but still evaluates mechanism specific scaling factors for photolysis of isoprene and terpene oxidation products.

CESM/CAM-chem TS1 base case model output was used to initialize BOXMOX. In order to pick a location with high

biogenic emissions, the grid box containing the Coffeeville U.S. EPA CASTNET monitoring site located in Mississippi (lat = 34.002747, lon = -89.799183) was selected because it has a forest land use type as classified by the U.S. EPA. Aug 3, 2013 was selected to represent this ideal day due to high biogenic emissions (i.e., highest noon isoprene and monoterpene emissions in August) and minimal cloud cover (i.e. within top five highest noon $j_{NO2}$ values in August). To reduce complexity and increase traceability, each box model simulation was initialized with only one non-oxygenated VOC at a time (e.g., only isoprene or

$\alpha$-pinene). To ensure reasonable oxidant concentrations, isoprene, monoterpene, and sesquiterpene emissions were scaled by 1.6, 13, and 88, respectively, such that their molar total was equal to the molar total emissions of the major non-oxygenated VOCs (alkanes, alkenes, aromatics, isoprene, and terpenes). This method was selected instead of holding the oxidants (i.e. OH, $O_3$, and $NO_3$) constant in order to evaluate differences in $O_3$, $HO_x$, and $NO_x$ between the mechanisms. These are ideal scenarios designed to examine and clearly present the impact variations in the chemistry of a single VOC have on oxidants and

oxidation products and not to accurately represent the chemistry of a specific location.

### 3.2 Global Modeling

The Community Earth System Model/Community Atmosphere Model with chemistry (CESM[TM]/CAM-chem) released version 2.1.0 was used with a horizontal resolution of 0.9° x 1.25° (Emmons et al., 2019; Tilmes et al., 2019). The meteorology (temperature and winds) was nudged with a 50 h relaxation time as described in Lamarque et al. (2012) to the Modern-

Era Retrospective analysis for Research and Applications, Version 2 (MERRA2) (Gelaro et al., 2017) meteorological fields interpolated to the native model resolution of 32 levels. CAM physics, dynamics, and cloud parameterizations are tuned at this 32 level vertical resolution and using a weak relaxation time (50 h) for nudging reduces variability while also limiting the impact of nudging on model parameterizations. In Figures S5 and S6, the impact of using different vertical resolutions and nudging relaxation times are shown for ozone, ozone precursors, and isoprene oxidation products. Using a stronger nudging

relaxation time (i.e 6 h rather than the 50 h used in this study) increases model bias in the vertical profile shape of ozone (Figure S5). Biogenic emissions were calculated online in the community land model (CLM) based on Model of Emissions of Gases and Aerosols from Nature (MEGAN) v2.1 (Guenther et al., 2012). Satellite derived plant functional type (PFT) and leaf area index (LAI) from AVHRR and MODIS data are used in the CLM model (Lawrence and Chase, 2007). The default biogenic emissions used in CESM/CAM-chem v2.1.0 were expanded to include more volatile organic compounds as listed in Table S3

for all simulations. Global anthropogenic emissions are from the Community Emissions Data System (CEDS) (Hoesly et al., 2018) and global biomass burning emissions are from van Marle et al. (2017).

Four main model simulations were conducted. First, the "TS1" case uses the default CESM2.1.0/CAM-chem code and the default TS1 chemical mechanism with an expansion of the biogenic volatile organic compounds emitted from the land model (Table S3) and using 32 vertical levels as described above. Second, the "Henry's Law" case uses the Henry's law constant





updates as described in Section 2.1 in the wet and dry deposition schemes. Third, the "ISOP" case uses the Henry's law updates and the isoprene oxidation chemistry described in Section 2.2. Fourth, the "TS2" case includes all of the TS2 chemistry updates: Henry's law, isoprene, and terpene updates (described in Section 2.3). Each case is progressively more complicated. Because new surrogates incorporated into the chemical mechanism of CESM are initialized to 0, these four cases were separately spun-

up for 2.5 years to ensure that all new species were equilibrated and all simulations were performed consistently. Sensitivity tests were also conducted to evaluate the impact of uncertainties remaining in the chemical mechanism on simulated surface ozone. All sensitivity tests were identical to the TS2 case with small variations in the chemistry as described in Section 4.6. These sensitivity tests were spun-up from exact restarts from the TS2 case for 3 months, which is sufficient because no new species were added.

## 4  Results and Discussion

Evaluation against more explicit schemes like the Caltech mechanism and MCM verifies that TS2 more accurately simulates isoprene (Section 4.1) and terpene (Section 4.2) chemistry compared to TS1. These comparisons verify that the current number of tracers and reactions in TS2 are sufficient to reasonably capture the isoprene and terpene chemistry represented by more explicit schemes. Comparisons with CASTNET monitoring observations (Section 4.3) and SEAC[4]Rs field campaign data

(Section 4.4) suggests that ozone, ozone precursors, and $NO_x$ reservoir species are generally better represented in TS2 than TS1. The differences in simulated ozone between the two mechanisms are largely caused by differences in formation and fate of organic nitrates, which is explained in Section 4.5. Although much is known about isoprene and terpene chemistry, lingering uncertainties remain. These uncertainties are evaluated in Section 4.6 demonstrating that further studies on terpene oxidation and isoprene and terpene derived organic nitrate loss processes, in particular aerosol uptake, are needed in order to reduce

uncertainties in simulated surface ozone.

### 4.1  Isoprene Evaluation Against Explicit Schemes

As described in detail in Section 3.1, CESM/CAM-chem TS1 base case model output from the grid box containing the Coffeeville U.S. EPA CASTNET monitoring site was used to initialize BOXMOX so that TS2 can be compared to explicit chemical mechanisms like RCIM (the Caltech mechanism) and MCM in an idealized diurnal cycle. As shown in Figure 3, in general

TS1, TS2, MCM, and RCIM agree fairly well for major oxidants and isoprene oxidation products. TS1 already included a good general structure for isoprene oxidation (Section 2.2), which is likely why ozone changes from TS1 to TS2 are moderate (Figure 3) at least under the single NO regime tested by the box model. There are large differences in some of the low-$NO_x$ oxidation products (ISOPOOH and IEPOX) and the isomer distribution of the 1st-generation hydroxy nitrates (ISOPN). TS1 produces a fixed yield of $\delta$- and $\beta$-hydroxy nitrates while TS2 similar to MCM and RCIM allows for the $\delta$- and $\beta$-hydroxy

nitrate distribution to vary based on the isoprene $RO_2$ lifetime (Section 2.2.1).

The overall isoprene $RO_2$ distribution impacts the distribution of 1st- (ISOPN) and later-generation organic nitrates (e.g., NOA, NO3CH2CHO, MACRN, and MVKN). As shown in Figure 3, these organic nitrates are more similar to the explicit

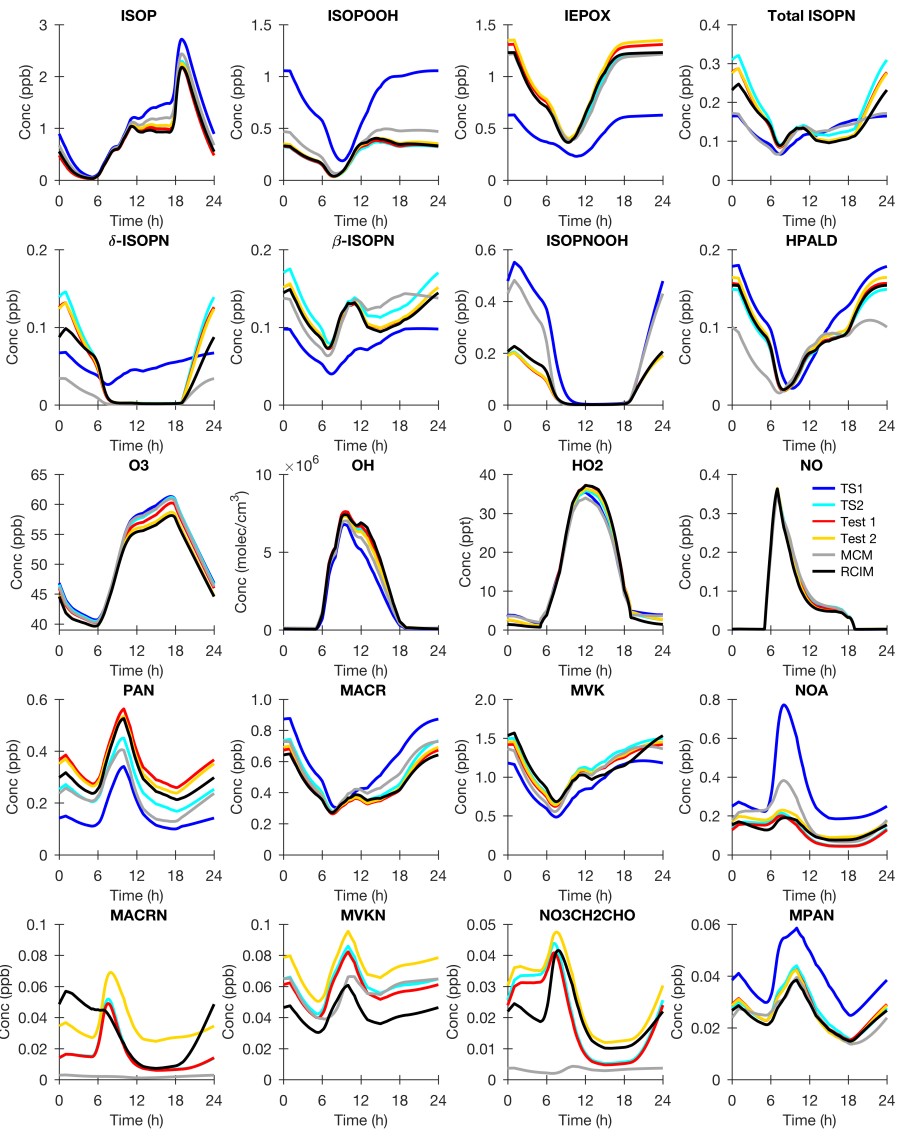

**Figure 3.** BOXMOX results for isoprene oxidation using TS1 (blue), MCM (gray), RCIM (black), TS2 (cyan), TS2 with RCIM assumptions for PAN and $C_4$ dihydroperoxy carbonyls - test 1 (red), and TS2 with RCIM assumptions for PAN, $C_4$ dihydroperoxy carbonyls, and carbonyl nitrates - test 2 (gold). The model configuration is explained in Section 3.1. The following acronyms are used: ISOP (isoprene), ISOPOOH (isoprene hydroxy hydroperoxide), IEPOX (isoprene dihydroxy epoxide), ISOPN (isoprene hydroxy nitrate), ISOPNOOH (isoprene nitrooxy hydroperoxide), HPALD (isoprene hydroperoxy aldehyde), O3 (ozone), OH (hydroxyl radical), HO2 (hydroperoxy radical), NO (nitrogen monoxide), PAN (peroxy acyl nitrate), MACR (methacrolein), MVK (methyl vinyl ketone), NOA (propanone nitrate), MACRN (methacrolein hydroxy nitrate), MVKN (methyl vinyl ketone hydroxy nitrate), NO3CH2CHO (ethanal nitrate), and MPAN (methacryloyl peroxynitrate).





schemes in TS2 than TS1. Simulated organic nitrates from $NO_3$ initiated oxidation are also improved. For example, in TS2 isoprene nitrooxy hydroperoxide (ISOPNOOH), which forms from isoprene + $NO_3$ oxidation, is consistent with the recently updated RCIM and not MCM or TS1 (Figure 3), because ISOPNOOH is no longer formed in unity yield in TS2 and RCIM (Section 2.2.3). In general, MCM over-estimates hydroperoxides because it consistently assumes unity yields when multi-

functional peroxy radicals react with $HO_2$ contrary to the most recent experimental evidence (Orlando and Tyndall, 2012; Wennberg et al., 2018).

As shown in Figure 3, RCIM simulates less ozone than TS2. Several sensitivity tests, which are explicitly listed in Table S7, were performed in order to understand this difference. In the first sensitivity test, TS2 was altered to use RCIM assumptions for formation and loss of PAN and for photolysis of $C_4$ dihydroperoxy carbonyls (DHPMPAL, Figure 1). In TS2, DHPMPAL

is added as a surrogate compound with fast, but not instantaneous photolysis rates. In RCIM, DHPMPAL is assumed to photolyze so fast that only its photolysis products are included in the mechanism. The PAN assumptions decrease $O_3$ and the DHPMPAL assumptions increase OH. Then for the second sensitivity test, TS2 was adjusted to include the assumptions in the first sensitivity test and the photolysis rates used in RCIM for carbonyl nitrates. This further reduces $O_3$ nearly to the level produced by RCIM itself (Figure 3).

TS2 was not updated based on these sensitivity tests. In TS2, reliable rate recommendations from JPL for PAN formation/thermal decomposition and OH oxidation are used (Burkholder et al., 2015). The photolysis of the $C_4$ dihydroperoxy carbonyls is unknown and expected to be fast, but possibly not instantaneous. Future studies measuring the photolysis rate of $C_4$ dihydroperoxy carbonyls are warranted given the impact on OH using different assumptions. The photolysis rates for the carbonyl nitrates in RCIM are lower than that suggested by recent experimental studies (Muller et al., 2014) and were recently

updated when RCIM was incorporated into the GEOS-Chem chemical transport model (Bates and Jacob, 2019).

In general, the box modeling results (Figure 3) demonstrate that TS2 simulates 1[st]- and later-generation isoprene oxidation products better than TS1 compared to explicit schemes. Confidence that simulated ozone is right for the right reasons is enhanced in TS2 because the chemistry is more accurately represented. These results confirm that although TS1 may have too few tracers to fully capture isoprene chemistry, adding only 39 more species significantly increases the chemical accuracy

without needing the immense cost that a nearly fully explicit chemical mechanism like MCM would require.

### 4.2  Terpene Evaluation Against Explicit Schemes

More significant changes were made to the terpene chemistry than the isoprene chemistry when developing TS2 (Section 2.3). TS2 is compared to TS1 and the more explicit MCM mechanism in Figures 4-6 and S3-S4, for all five surrogate compounds: APIN, BPIN, LIMON, MYRC, and BCARY. TS2 separates hydroxy nitrates and hydroxy hydroperoxides based on their

chemical structure (i.e. primary/secondary versus tertiary, saturated or unsaturated, and presence of multi-functional groups) instead of their generation. Thus, MCM $C_4$ and greater organic hydroperoxides, nitrates, and peroxy acyl nitrates are summed in Figure 4 as a fairer comparison with the surrogate compounds in TS2. Nitrooxy hydroperoxides (NTERPOOH) are not included as total nitrates or total hydroperoxides because TS2 considers these separately.

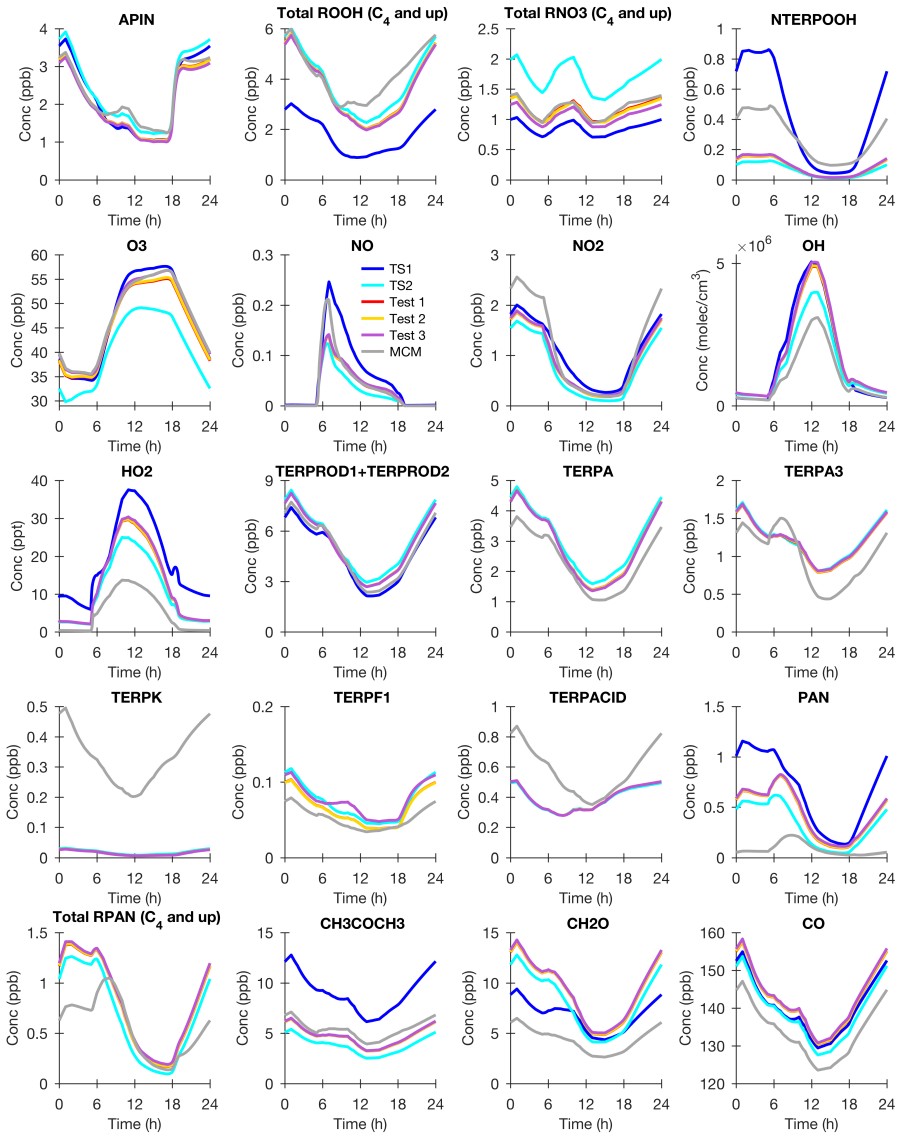

**Figure 4.** BOXMOX results for $\alpha$-pinene (APIN) oxidation using TS1 (blue), MCM (gray), TS2 (cyan), TS2 with MCM pinonaldehyde nitrate yield - Test 1 (red), TS2 with MCM pinonaldehyde and limonaldehyde nitrate yield - Test 2 (gold), and TS2 with MCM pinonaldehyde and limonaldehyde nitrate yield and assumptions for oxidation of unsaturated hydroxy nitrates - Test 3 (purple). The model configuration is explained in Section 3.1. APIN ($\alpha$-pinene surrogate), TERPOOH (terpene hydroxy hydroperoxide), Total ROOH (all terpene hydroperoxides $C_4$ and up), TERPNIT (terpene hydroxy nitrate), Total RNO3 (all terpene nitrates $C_4$ and up), NTERPOOH (terpene nitrooxy hydroperoxide), CH3COCH3 (acetone), TERPROD1 + TERPROD2 (all terpene $1^{st}$ + $2^{nd}$ generation products except hydroperoxides, nitrates, and PANs), TERPA (terpene aldehyde like pinonaldehyde), TERPA3 (terpene aldehyde like limonaldehyde), TERPK (terpene ketone), TERPF1 (terpene product - one double bond), TERPACID (terpene acid), PAN (peroxy acyl nitrate), and Total RPAN (all terpene PANs $C_4$ and up).





TERPROD1 and TERPROD2 are terpene oxidation products from 1$^{st}$- and 2$^{nd}$-generation chemistry respectively in TS1. In TS2, these have been separated based not on their generation, but on their chemical structure: terpene aldehydes (TERPA, TERPA2, TERPA3), terpene ketones (TERPK), terpene unsaturated products (TERPF1, TERPF2), and terpene acids (TERPACID, TERPACID2, TERPACID3) (Section 2.3.1). MCM 1$^{st}$ and 2$^{nd}$-generation species are combined into the same cate-

gories for comparison with the TS2 mechanism. In general, the total products produced are similar and the types of compounds formed are reasonably consistent between MCM and TS2.

Although MCM is one of the most explicit chemical mechanisms available, after careful examination of the chemistry, there are a number of general assumptions in MCM that are outdated or overly simplified for terpene oxidation. Even though TS2 is more condensed compared to MCM, TS2 may be more accurate because it has been updated more recently and simplifications

used in TS2 were carefully selected to limit their impact on $NO_x$ and $HO_x$ recycling.

To justify this, several sensitivity tests were performed as summarized in Table S7. These sensitivity tests confirm that much of the disagreement between MCM and TS2 is due to several differences in assumptions and not due to the simplification process. The first sensitivity test reduced the nitrate yields from pinonaldehyde oxidation to those recommended by MCM. TS2 uses the nitrate yield estimation procedure in Wennberg et al. (2018) with an upper limit of 0.3 as more explicitly described in

Section 2.3.1. The second sensitivity test included adjustments from the first sensitivity test and reduced the nitrate yields from limonaldehyde oxidation to those recommended by MCM. The third sensitivity test included all the adjustments in the second sensitivity test as well as adjustments in the MCM assumptions for the oxidation of unsaturated hydroxy nitrates. Commonly, in MCM, unsaturated hydroxy nitrates will react with OH via H-abstraction to produce only $NO_2$ and an unsaturated aldehyde or ketone rather than via addition to the double bond to produce a multi-functional organic nitrate. Because OH addition to

a double bond consistently occurs at a faster rate than H-abstraction, this simplification inaccurately increases $NO_2$ recycling and ozone formation.

Similarly, in MCM unsaturated hydroxy hydroperoxides will react with OH via H-abstraction to produce OH and an unsaturated aldehyde or ketone or via H-abstraction of the hydrogen on the hydroperoxide group to form a hydroxy peroxy radical. In particular, when OH reacts with an unsaturated hydroperoxide with a fast rate, given the presence of a double bond, to H-

abstract the hydrogen on the hydroperoxide moiety, an unrealistically fast cycle is created (i.e., $RO_2 \xrightleftharpoons[OH]{HO2} ROOH$; each cycle removes 2 $HO_x$ radicals). These unrealistically fast cycles are commonly used in MCM to reduce the number of species and can lead to removal of $HO_x$ depending on the number of cycles that occur, overestimation of the 1$^{st}$-generation hydroperoxides, and under-estimation of later-generation low-volatility hydroperoxides, which will impact predicted SOA if SOA formation is coupled directly to the gas-phase chemistry.

Unlike MCM, TS2 does not contain these unrealistically fast cycles and is designed to accurately account for low-volatility compounds, so that in the future terpene SOA formation can be accurately coupled directly to gas-phase chemistry like recent studies have done for isoprene (Marais et al., 2016; Bates and Jacob, 2019; Stadtler et al., 2018) and monoterpenes (Zare et al., 2019). In MCM, unsaturated aldehydes and ketones typically react with OH via addition to the double bond as well as H-abstraction, so this simplification only impacts unsaturated hydroxy nitrate and hydroxy hydroperoxide oxidation. Future



studies that use MCM for prediction of ozone or SOA in terpene rich regimes should update these simplifications, so that ozone, SOA, and the $HO_x$ and $NO_x$ budgets are accurately simulated.

In Figure 4, the $\alpha$-pinene (APIN) surrogate compound is compared with TS1 and MCM. Terpene oxidation is largely based on $\alpha$-pinene in TS1, so as expected APIN in TS2 compares reasonably well with MTERP in TS1 (Figure 4). When accounting for total hydroperoxides, MCM and TS2 agree well. Total organic nitrates are higher and ozone is lower in TS2 than MCM mostly due to the use of different nitrate yields from pinonaldehyde oxidation (sensitivity test 1). Unfortunately, there are few studies measuring the organic nitrate yields from later-generation products, making it difficult to determine whether the organic nitrate yield parameterizations from MCM are better than those from Wennberg et al. (2018) used in TS2. These sensitivity tests demonstrate that more measurements of organic nitrate yields from multi-functional later-generation products are needed. NTERPOOH is larger in MCM than TS1 because MCM in general assumes a unity yield of hydroperoxides from all multi-functional peroxy radicals reaction with $HO_2$ contrary to recent studies (e.g., Kurten et al., 2017).

Because the $\beta$-pinene (BPIN) surrogate compound is less reactive with $O_3$ and $NO_3$ than $\alpha$-pinene, TS1 greatly under-estimates the $\beta$-pinene concentration at night and does not simulate $\beta$-pinene oxidation products well either (Figure 5). In TS1, organic nitrates are under-predicted and ozone is over-predicted compared to MCM. There are some differences in MCM compared to TS2. The organic nitrates are larger in the TS2, but most of this bias is caused by inaccurate oxidation of the unsaturated hydroxy nitrates in MCM (sensitivity test 3). The nopinone yield (TERPK) for OH- and $NO_3$-initiated oxidation of $\beta$-pinene in TS2, which is based on recent experimental and theoretical studies (Section 2.3.1 and 2.3.3), is much lower than the yield assumed by MCM. The TS2 mechanism assumes the formation of aldehyde type products based largely on theoretical studies (Vereecken and Peeters, 2012; Kurten et al., 2017) and consistent with a recent experimental study as well (Xu et al., 2019). Most reduced chemical schemes do not separate $\alpha$-pinene and $\beta$-pinene, but the results from these box model comparisons demonstrate the need for such separation (Figure 4 and 5). TS1, which largely assumes all monoterpenes react like $\alpha$-pinene, simulates very different oxidant and terpene oxidation product concentrations than TS2.

All limonene (LIMON) surrogate compounds contain two double bonds and so are more reactive than $\alpha$-pinene. Because TS1 does not account for this extra double bond, limonene oxidation is not well represented compared to MCM (Figure 6). As explained above, contrary to experimental knowledge, MCM assumes that OH reacts with an unsaturated hydroxy nitrate to hydrogen abstract rather than react with a double bond (sensitivity test 3), which causes a large underprediction of organic nitrates and an overprediction of ozone. This especially impacts limonene oxidation, which due to the presence of two double bonds, produces unsaturated first-generation products. Consistent with MCM, limonene oxidation in TS2 produces mostly terpene unsaturated products like limonaldehyde and limaketone (TERPF1). In MCM, the nitrate yields from these terpene unsaturated products are lower than that assumed in TS2, which has a moderate impact on the total organic nitrates and ozone (sensitivity test 2). TS2 simplifies limonene chemistry by not explicitly tracking aldehydes on hydroperoxides, nitrates, or unsaturated terpene products and by not adding separate tracers for unsaturated and saturated terpene acids and terpene PANs. These simplification cause some disagreement between TS2 TERPF1, TERPA3, TERPACID, and total terpene PANs compared to MCM. Even with these simplifications, TS2 terpene PANs, which will impact the $NO_x$ budget, are consistent with MCM. At this time, later-generation chemistry from limonene oxidation is not understood well enough to motivate increasing

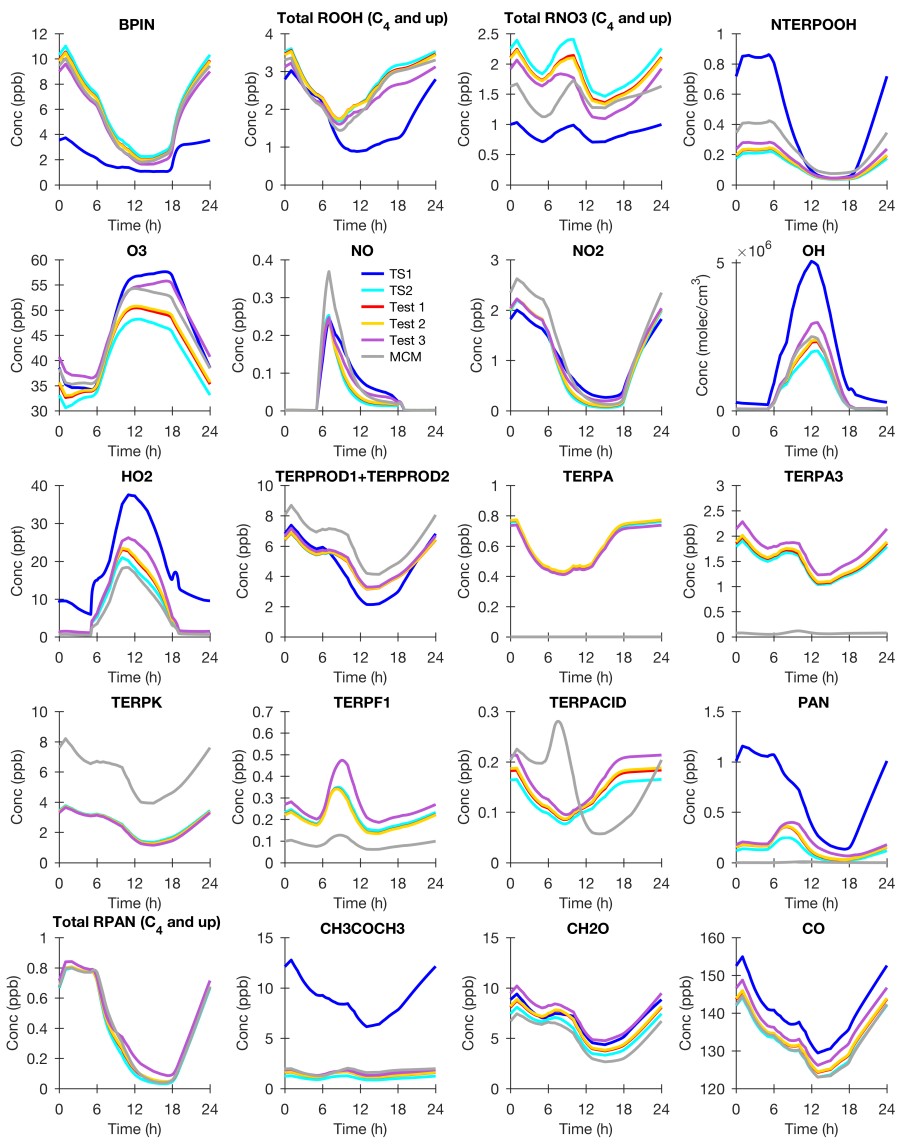

**Figure 5.** BOXMOX results for $\beta$-pinene (BPIN) oxidation using TS1 (blue), MCM (gray), TS2 (cyan), TS2 with MCM pinonaldehyde nitrate yield - Test 1 (red), TS2 with MCM pinonaldehyde and limonaldehyde nitrate yield - Test 2 (gold), and TS2 with MCM pinonaldehyde and limonaldehyde nitrate yield and assumptions for oxidation of unsaturated hydroxy nitrates - Test 3 (purple). The model configuration is explained in Section 3.1. All species names are identical to Figure 4.

the computational cost by adding in many additional tracers to directly track the presence of aldehydes along with the other functional groups (e.g., nitrates, hydroperoxides, unsaturated products).

The box model comparisons for $\beta$-caryophyllene (BCARY) and myrcene (MYRC) surrogate compounds are shown in Figure S3 and S4. Because these surrogate compounds contain more than one double bond, the differences between TS2, TS1, and





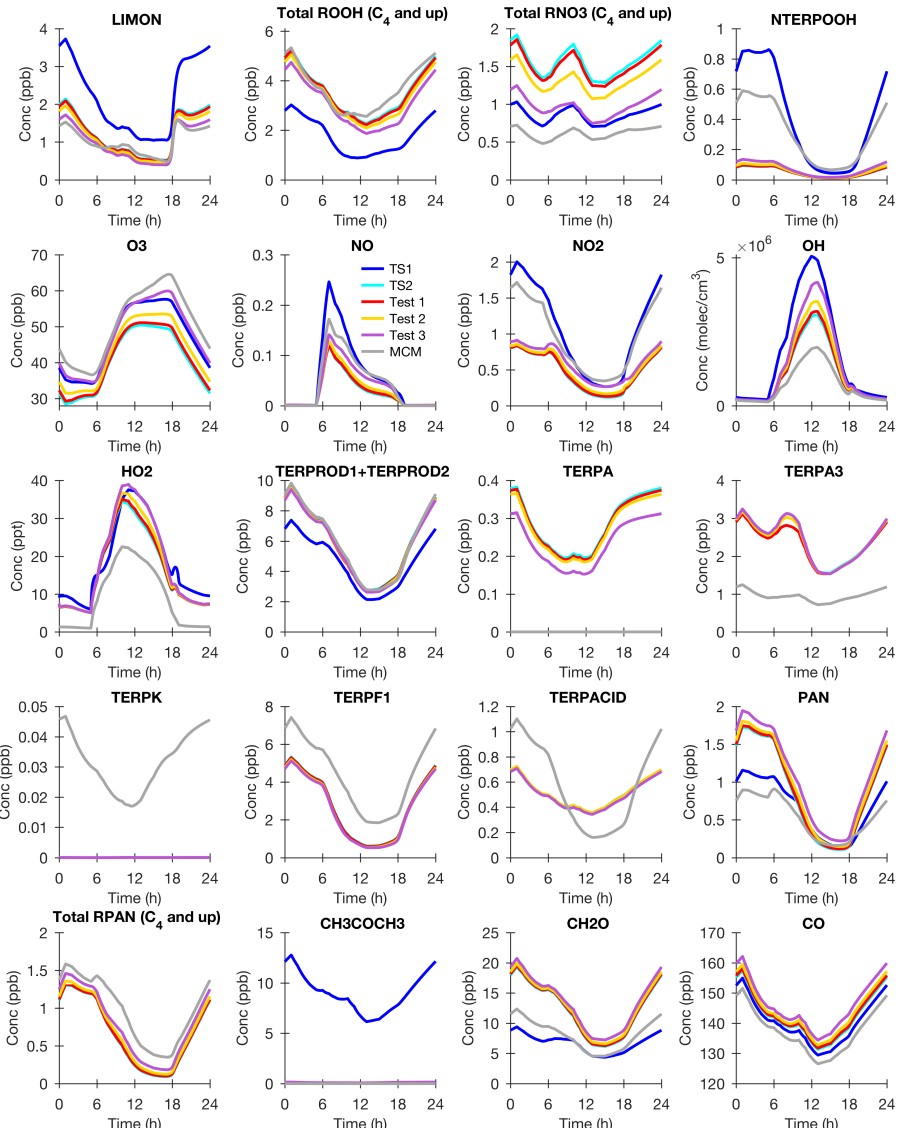

**Figure 6.** BOXMOX results for limonene (LIMON) oxidation using TS1 (blue), MCM (gray), TS2 (cyan), TS2 with MCM pinonaldehyde nitrate yield - Test 1 (red), TS2 with MCM pinonaldehyde and limonaldehyde nitrate yield - Test 2 (gold), and TS2 with MCM pinonaldehyde and limonaldehyde nitrate yield and assumptions for oxidation of unsaturated hydroxy nitrates - Test 3 (purple). The model configuration is explained in Section 3.1. All species names are identical to Figure 4.

MCM are similar to those shown for limonene in Figure 6. Comparisons against MCM for myrcene are not available because no equivalent tracer exists in MCM. The TS2 surrogate compound for $\beta$-caryophyllene is assumed to have three double bonds since sesquiterpenes have between 2 and 4 double bonds (Guenther et al., 2012), so differences are expected when comparing directly with MCM's $\beta$-caryophyllene, which only has two double bonds.





From all of the box model comparisons (Figures 4-6 and S3-S4), some general trends for terpene oxidation can be deduced. As explained above, MCM v3.3.1 has some limitations that influence $O_3$ formation. Especially for reactive terpenes with more than one double bond, MCM does not accurately represent $NO_x$ recycling because of inaccurate simplifications for unsaturated hydroxy nitrate oxidation. Because this is an erroneous representation of the chemistry, this sensitivity test is not evaluated in the global model in Section 4.6. The pinonaldehyde and limonaldehyde nitrate yields assumed in MCM are lower than those used in the TS2 mechanism (Sensitivity test 1 and 2). The nitrate yields from these later-generation products have not been experimentally measured and are highly uncertain. The impact of assuming lower nitrate yields from these 1st-generation terpene oxidation products are tested in the global model in Section 4.6. Using a more complex terpene oxidation scheme like TS2 seems to be important for accurately simulating later-generation products in addition to oxidants and 1st-generation products. For example, TS1 consistently over-estimates the formation of acetone for all of the terpene surrogate compounds compared to MCM and TS2.

### 4.3 Surface Ozone Impact in CESM/CAM-chem

Comparing models with surface ozone measurements from the U.S. EPA CASTNET monitoring network is particularly useful compared to air quality monitoring data because typically stations in the CASTNET network are in rural locations and carefully selected to be representative of a specific region (U.S.EPA). As discussed in the introduction, CESM/CAM-chem using TS1 chemistry substantially overestimates surface ozone during the summer in the eastern U.S. compared to data from the CASTNET monitoring network (Figure 7a). Consistent with the box model results shown in Sections 4.1 and 4.2, the ozone bias is greatly improved with the updated TS2 mechanism, but a large bias remains (Figure 7b). In all cases, the surface ozone is the value in the lowest model layer with a midpoint of $\sim 66$ m above the ground. Brown-Steiner et al. (2015) determined correcting CESM/CAM-chem data to 10 m where CASTNET data are measured reduced MDA8 surface ozone by $\sim 2\%$. For now no correction was applied, but future work with more comprehensive comparisons to CASTNET monitoring data will evaluate this correction.

The chemistry updates were added in sequential order so that the effect of each update on ozone could be diagnosed. As shown in Figure 8a, updates to the Henry's law constants had only moderate effects on MD8A surface ozone, but in certain locations reduced MDA8 ozone by a couple ppb. Updates to Henry's law constants and isoprene reduced MDA8 ozone more consistently throughout the U.S. by a couple ppb and up to 6 ppb (Figure 8b). All TS2 updates (Henry's law, isoprene, and terpene updates) reduced MDA8 ozone generally in the eastern U.S. by around 4-5 ppb and up to 7 ppb (Figure 8c). In particular, the terpene updates reduced the MDA8 ozone bias most substantially in the southeast U.S. where the model ozone bias is largest (Figure 7). Terpene chemistry greatly impacts simulated surface ozone even though it has received much less attention in model and experimental studies in the past.

### 4.4 Evaluation Against Field Campaign Data

The CASTNET monitoring data is useful for evaluating surface ozone itself. However, to verify that the model is accurately simulating ozone for the right reasons, evaluation against ozone precursors (e.g., $NO_x$ and VOCs) and $NO_x$ reservoir species



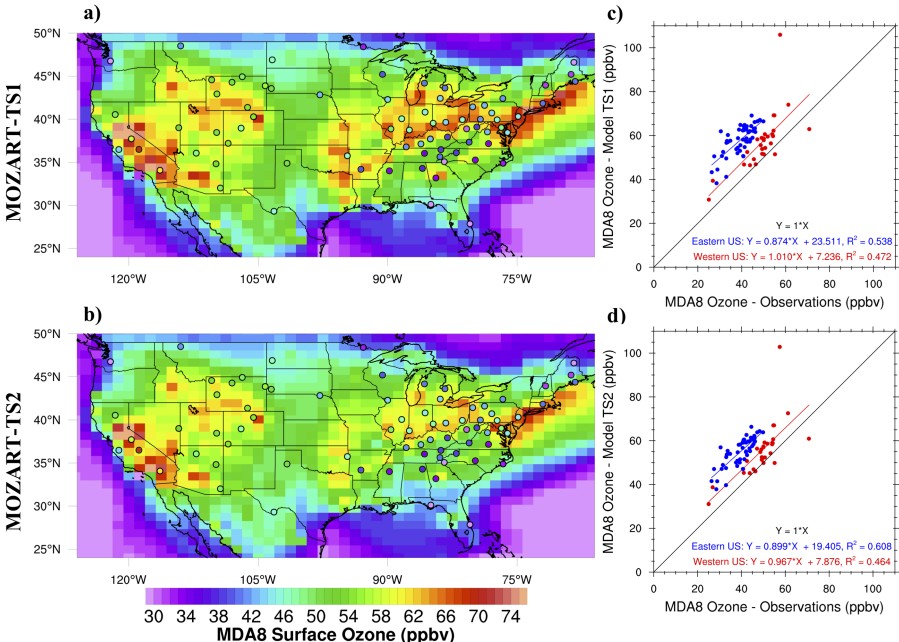

**Figure 7.** Surface Ozone Daily Max 8-hr Average (MDA8) CESM/CAM-chem results over CONUS using the default TS1 mechanism (panel a) and the updated TS2 mechanism (panel b) compared to U.S. EPA CASTNET data (filled circles) averaged over August 2013. The same data are presented in a scatter plot for TS1 (panel c) and TS2 (panel d) with Eastern U.S. (longitude > -96°) in blue and Western U.S. (longitude < -96°) in red.

(e.g., PANs and organic nitrates) is necessary. Field campaigns like Studies of Emissions and Atmospheric Composition, Clouds and Climate Coupling by Regional Surveys (SEAC[4]RS), whose goals included investigating oxidation chemistry of biogenic VOCs and measuring ozone, ozone precursors, and $NO_x$ reservoir species, are critical for evaluating whether models are accurately representing ozone. The SEAC[4]RS field campaign was conducted during August through September of 2013

5 (Toon et al., 2016) with many of the flights centered over the southeastern U.S. In SEAC[4]RS and in many other past field campaigns, terpene oxidation products were not quantitatively measured. More measurements in the future of terpene oxidation products will be beneficial for further evaluation of TS2.

Median vertical profiles are displayed in Figure 9 where data are grouped into 0.5 km bins and urban plumes ($NO_2 > 4$ ppb), fire plumes (acetonitrile > 0.2 ppb), and stratospheric air ($O_3/CO > 1.25$) are excluded as done in previous work (Travis et al.,

10 2016). Only the southeastern U.S. was selected for local sun times between 9 am to 5 pm. Version 7 of the SEAC[4]4RS 60 s merge was used and data flagged as missing or as an upper limit of detection were not used and data flagged as a lower limit of detection were set to 0. For each compound, data unavailable in the observational dataset were also removed from the model dataset. This ensures that the observational and modeling data for each compound are directly comparable, but in some cases also prohibits direct comparison between different species since the sampling data may be different. For example, the model





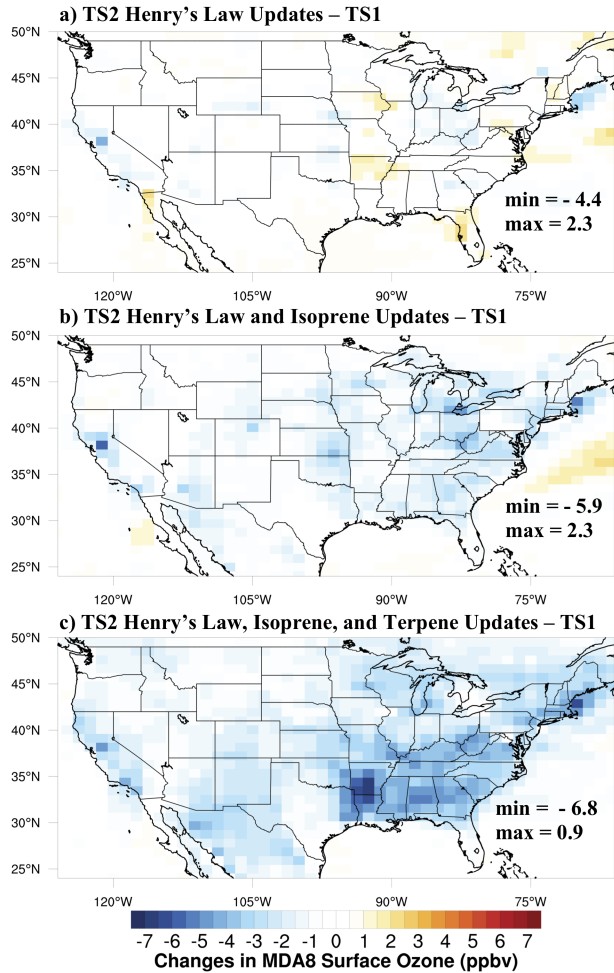

**Figure 8.** Changes in surface ozone daily max 8-hr average (MDA8) between the TS2 updated case and the TS1 case averaged over August 2013. The minimum and maximum over the domain pictured is displayed on the bottom right for each case.

vertical profiles for $NO_2$ are different for the chemilumenescence instrument from NOAA Earth System Research Laboratory (ESRL) and the thermal dissociation-laser induced fluorescence (TD-LIF) instrument because each instrument has different unavailable data (Figure 9). As shown in Figure 9, even though CESM/CAM-chem was only nudged to meteorological data with a 50 h relaxation time, temperature (T), winds (zonal - U and meridional -V), and clouds (as evaluated by $NO_2$ photolysis) were all consistent between the simulations ensuring that this light nudging sufficiently reduced model variability.

Consistent with the surface ozone analysis in Section 4.3, CESM/CAM-chem over-predicts ozone throughout the planetary boundary layer (PBL). The Henry's law and isoprene updates do not change ozone much above the surface while the terpene updates do reduce ozone in the PBL. The vertical profile shapes for ozone, NO, isoprene, and monoterpenes are quite different between the model and observations suggesting that updates to the PBL height, mixing schemes, clouds (Ryu et al., 2018),

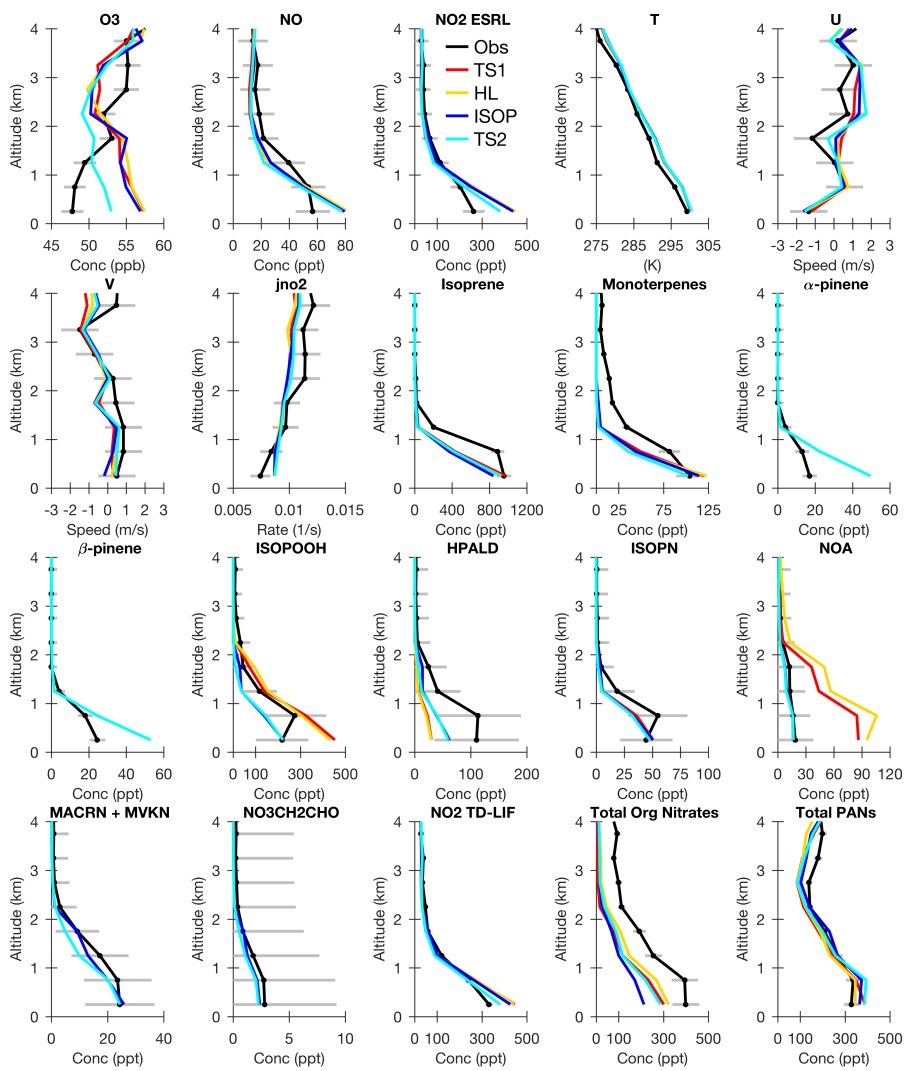

**Figure 9.** Median vertical profile plots over the SEAC[4]Rs flight tracks for observations (Obs - black), TS1 (red), TS2 with only Henry's Law updates (HL - gold), TS2 with Henry's Law and isoprene updates (ISOP - blue), and TS2 with Henry's Law, isoprene and terpene updates (TS2 - cyan). Acronyms are defined in Figure 3. Data are grouped into 0.5 km bins and exclude urban plumes ($NO_2 > 4$ ppb), fire plumes (acetonitrile > 0.2 ppb), and stratospheric air ($O_3/CO > 1.25$) as done in previous work (Travis et al., 2016). Domain includes the southeast U.S. (29.5-40°N, 75-94.5°W), and local sun time 9 am to 5 pm. Observational uncertainty is shown in gray bars.

vertical resolution, or ozone dry deposition schemes (Clifton et al., 2019) may be needed to further reduce ozone biases in CESM/CAM-chem.

Although the simulated $NO_2$ photolysis ($j_{NO2}$ - Figure 9) is within the uncertainty of the observations, biases within the $NO_2$ photolysis vertical profile shape suggest biases in simulated clouds. Sampling biases could also impact agreement between the





model and observations for $NO_2$ photolysis because field campaigns typically avoid sampling clouds on a scale not resolved by the model (Hall et al., 2018). In Ryu et al. (2018), WRF-chem similar to CAM-chem under-predicts $NO_2$ photolysis above 2 km and when clouds derived from satellites are incorporated into WRF-chem the $NO_2$ photolysis bias is removed and MDA8 surface ozone decreases by 1 to 5 ppb. Additionally, studies using large eddy simulations have determined that shallow

cumulus clouds enhance vertical transport of passive (i.e., no aqueous-phase processing) chemical compounds as compared to clear sky conditions (Vila-Guerau de Arellano et al., 2005; Li et al., 2017). The vertical biases in $NO_2$ photolysis combined with the biogenic VOCs (isoprene and monoterpenes) not adequately lofting above the surface in the model compared to the observations (Figure 9) possibly suggest that CAM-chem is not accurately simulating the dynamics and/or location of shallow cumulus clouds. Cloud biases in CAM-chem will be further evaluated in future work due to their potential to improve simulated

$NO_2$ photolysis, biogenic VOCs, and ozone in and above the PBL.

NO and $NO_2$, as measured by ESRL using chemilumenescence, are over-predicted near the surface in CESM/CAM-chem for all model simulations. However, $NO_2$ is reduced with the updates to the terpene chemistry. This demonstrates that disagreement between simulated and observed $NO_2$ is not necessarily due to emissions, but can also be caused by an inaccurate representation of the losses of $NO_x$ or $NO_x$ reservoir compounds. Interestingly, the model bias compared to measurements for $NO_2$ from

the TD-LIF instrument is lower than that from ESRL, which will be explored more in future simulations at finer horizontal resolution where $NO_x$ emissions are better resolved.

In general, biogenic VOCs (isoprene and monoterpenes) are under-predicted by CESM/CAM-chem compared to the SEAC[4]Rs observations (Figure 9). Even though the total monoterpenes measured using a proton transfer reaction - mass spectrometer (PTR-MS) are under-predicted, interestingly $\alpha$-pinene and $\beta$-pinene measured by the whole air sampler are over-predicted by

CESM/CAM-chem near the surface when compared to the APIN and BPIN surrogate compounds, respectively. The APIN surrogate compound largely only includes $\alpha$-pinene while the BPIN surrogate compound includes $\beta$-pinene and many other monoterpenes with a single double bond (Table S3). At least for $\beta$-pinene, possibly the unmeasured other monoterpenes included in the BPIN surrogate compound contribute to this bias. Further analysis at higher horizontal resolution where BVOC emissions are better resolved will be done in the future to explore the differences between CESM/CAM-chem and the obser-

vations and also the differences between the two monoterpene measurement techniques.

The slight under-prediction in the biogenic VOCs by CESM/CAM-chem could be caused by a number of possibilities. The $0.9°\text{x}1.25°$ resolution with a 30 minute time step may not be sufficient to capture the varied landscape in the southeast U.S., which has a number of large urban regions surrounded by forests. Additionally, Kim et al. (2016) used large eddy simulations to demonstrate that isoprene chemical reactivity is impacted by turbulent mixing in the boundary layer because these two

processes occur at similar rates. Considering the model bias in isoprene and monoterpenes is particularly high in the middle of the PBL, this effect may explain part of the low bias for isoprene and monoterpenes with similar reactivities to isoprene (e.g., limonene). However, the isoprene 1[st]-generation oxidation products, which are less reactive than isoprene and so less impacted by turbulent mixing, are also low compared to the SEAC[4]Rs observations. This then suggests that emissions may be simply biased low for isoprene and monoterpenes. Interestingly, GEOS-Chem using the same MEGAN v2.1 algorithm as used in

CESM/CAM-chem, recently concluded that isoprene emissions are too high by 40% in the southeast U.S. (Kaiser et al., 2018).



Although the emissions factors are likely the same in the two models, the surface temperatures are different and CESM/CAM-chem uses LAI and plant functional types from the community land model (CLM). All of the discussed possibilities may play some role in the bias warranting a more complete study evaluating biogenic emissions in CESM/CAM-chem in the future.

TS2 updated isoprene chemistry generally improves the description of isoprene 1st- and later-generation oxidation products (Figure 9). Because isoprene is under-predicted in CESM/CAM-chem, as expected the isoprene 1st-generation products are also under-predicted. The high bias in isoprene hydroxy hydroperoxide (ISOPOOH), which forms under low-NO conditions, is corrected in TS2 relative to TS1. HPALD, which includes all four isomers of isoprene hydroperoxy aldehydes as well as the isoprene carbonyl hydroxy epoxide, is closer to the observations in TS2 than TS1, but still under-predicted. HPALD yields and photolysis/oxidation are rather uncertain. Additionally, HPALD formation, which occurs under very low-NO levels from peroxy radical H-shifts (Figure 1), may be quite sensitive to horizontal resolution. For example, large grid boxes with averaged medium levels of $NO_x$ will produce different HPALD production rates than smaller grid boxes with separated high and low levels of $NO_x$. HPALD formation will be further evaluated in future work at finer horizontal resolution. In TS2, the formation and loss processes of isoprene hydroxy nitrates (ISOPN) are both increased resulting in very little overall difference in ISOPN concentration even though there are clear differences in the chemistry. TS1 groups all later-generation organic nitrates as propanone nitrate (NOA). The addition of more tracers in TS2 improves the description of these later-generation organic nitrates (NOA, MACRN+MVKN, and NO3CH2CHO) when compared to observations. Grouping all later-generation isoprene organic nitrates as NOA produces too many later-generation organic nitrates as the fate of NOA is different from the fate of the newly added tracers (MACRN+MVKN, and NO3CH2CHO). More measurements of terpene oxidation products in future field campaigns would be useful for a similar evaluation of reduced terpene mechanisms.

Total organic nitrates are under-predicted in all simulations compared to observations, while peroxy nitrates are slightly over-predicted. If the ultimate fate of these missing organic nitrates is to remove $NO_x$, this could explain part of the remaining ozone bias. The low model bias of total organic nitrates is concerning as this implies that the $NO_x$ budget is not completely understood. Part of this bias is caused by not including organic nitrates from small chain ($C_3$ and under) alkanes and alkenes as demonstrated by Fisher et al. (2016). Although the organic nitrate yields are lower for smaller chain alkanes and alkenes (Arey et al., 2001; Butkovskaya et al., 2010, 2012; Teng et al., 2015), these alkyl nitrates are still important in the atmosphere (Teng et al., 2015). Alkane and alkene nitrates and resulting chemistry will be added to the model in the future, but will only explain a small part of this model bias (Fisher et al., 2016). The overall low model bias for total organic nitrates is consistent with past work using other models (Fisher et al., 2016; Li et al., 2018). CESM/CAM-chem with the TS2 scheme produces less of a bias for total organic nitrates than GEOS-Chem (Fisher et al., 2016) likely due to the improved terpene chemistry as further explained in Section 4.5.

## 4.5 Organic Nitrate Formation and Fate

Because the total organic nitrates for all simulations are under-predicted compared to the SEAC$^4$RS observations (Figure 9), the speciation of organic nitrates in TS1 versus TS2 over the SEAC$^4$RS flight tracks are compared demonstrating that large differences in isoprene and terpene derived organic nitrates exist between the two mechanisms (Figure 10). For isoprene derived




organic nitrates, TS2 clearly has more $\beta$- than $\delta$-isoprene hydroxy nitrates consistent with recent experimental and theoretical studies (Teng et al., 2017; Peeters et al., 2014) (Figure 10a and b). The total isoprene derived organic nitrates are lower in TS2 than TS1 largely due to the more explicit treatment of later-generation organic nitrates in TS2. In TS1, all later-generation organic nitrates are grouped as NOA. Additional surrogate compounds are added in TS2 including MACRN, MVKN, and

5    NO3CH2CHO, which have higher photolysis rates than NOA. The faster loss processes lead to a lower total organic nitrate level and faster $NO_x$ recycling in TS2 versus TS1. The model comparisons against SEAC$^4$RS observations (Figure 9) confirm, that the representation of later-generation isoprene organic nitrates in TS2 is more accurate than TS1.

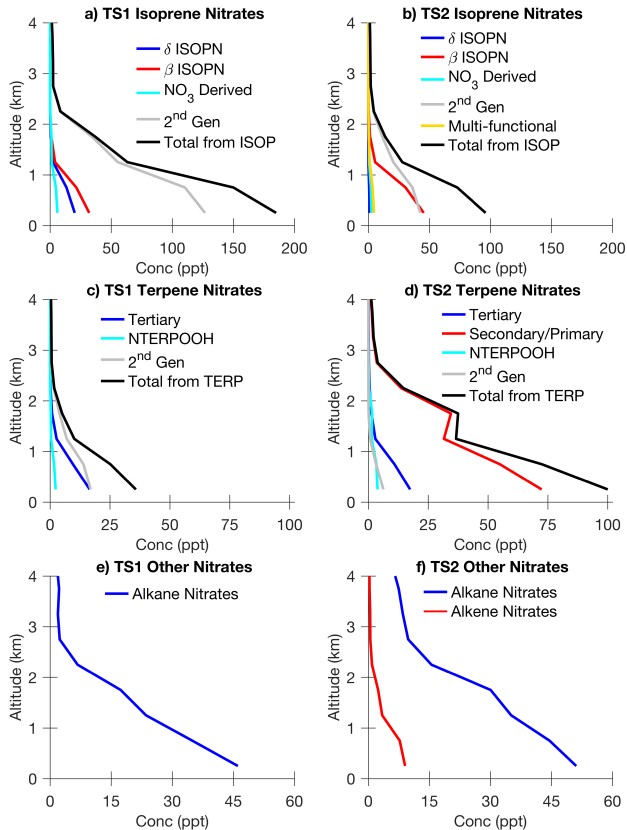

**Figure 10.** Median vertical profile plots over the SEAC$^4$Rs flight tracks for all isoprene, terpene, and other nitrates formed in the TS1 and TS2 mechanisms binned as specified in Figure 9. For isoprene, ISOPN includes all isoprene hydroxy nitrates, $NO_3$ derived includes all nitrates derived only from $NO_3$-initiated oxidation (i.e. ISOPNOOH, INHE, and NC4CHO - Figure S1), 2$^{nd}$ Gen includes all 2$^{nd}$ generation nitrates, and multi-functional includes all multi-functional later-generation nitrates. For terpenes, tertiary includes all tertiary hydroxy nitrates, secondary/primary includes all secondary/primary hydroxy nitrates, NTERPOOH includes all tertiary hydroperoxy nitrates, and 2$^{nd}$ Gen includes all 2$^{nd}$ generation nitrates, which for TS2 includes only the multi-functional low-volatility organic nitrates (Section 4.2). All dinitrates are doubled.





In contrast, terpene derived organic nitrates are higher in TS2 than TS1 (Figure 10c and d), which is largely due to the separation of tertiary and primary/secondary organic nitrates in TS2. Tertiary organic nitrates will undergo aerosol uptake and rapid hydrolysis to nitric acid while primary/secondary organic nitrates will not (Jacobs et al., 2014; Hu et al., 2011; Darer et al., 2011). In TS1, terpene derived hydroxy nitrates undergo aerosol uptake slower than TS2, but all hydroxy nitrates

undergo aerosol uptake (Section 2.4). In comparison, TS2 has a higher uptake rate, but only tertiary hydroxy nitrates undergo aerosol uptake. Thus, in TS2, primary/secondary organic nitrates remain in the gas phase to be lost by other processes such as reaction with OH and photolysis. Consistent with other recent studies (Muller et al., 2018; Bates and Jacob, 2019; Zare et al., 2018), separating terpene tertiary organic nitrates from primary/secondary organic nitrates is clearly important for accurately representing organic nitrate concentrations and fate in the atmosphere (Figure 10).

Most of the organic nitrates in TS2 are from isoprene and terpene chemistry. The other organic nitrates (Figure 10e and f) are derived from alkanes and alkenes. Some isoprene derived nitrates are grouped with alkene nitrates in TS1. By adding in additional tracers into TS2 (MACRN and MVKN), the alkene nitrates are largely separated in TS2 and so can be evaluated independently. Even though isoprene and terpene emissions dominate, alkane and alkene nitrates are fairly important over the southeast U.S. in their contribution to the total organic nitrates (Figure 10) likely due to their relatively long atmospheric

lifetime.

In order to accurately simulate ozone for the right reasons in any model, the fate of the organic nitrates must be accurately described. As $NO_x$ levels decrease, mixed regimes are becoming more prevalent. For example, during the Southern Oxidant and Aerosol Study (SOAS), isoprene dihydroxy hydroperoxide nitrates were measured, which are likely derived from 1[st]-generation isoprene hydroxy nitrates reacting with OH to form a peroxy radical that reacts with $HO_2$ (Lee et al., 2016; Xiong

et al., 2015). TS1 fails to capture this mixed regime chemistry because when 1[st]-generation unsaturated organic nitrates react with OH the resulting peroxy radical is assumed to immediately react with NO to form stable products. In contrast, TS2 assumes 1[st]-generation unsaturated isoprene and terpene organic nitrates react with OH to form peroxy radicals that then react with NO or $HO_2$ or in the case of isoprene to isomerize. As an example in Figure 11, the dominant fate of the peroxy radicals ($ISOPNO_2$) formed from isoprene hydroxy nitrates reaction with OH in TS2 is to isomerize. For peroxy radicals ($TERPNO_2$)

formed from terpene unsaturated hydroxy nitrates + OH the dominant fate is reaction with NO, but $HO_2$ is still relevant across the U.S. Not constraining the fate of the peroxy radicals to the first generation fate as done in TS2 will be increasingly important as $NO_x$ levels decrease in the future.

The possible fates for organic nitrates in the atmosphere include reaction with OH or $O_3$, photolysis, wet/dry deposition, or aerosol uptake. Typically photolysis and reaction with OH or $O_3$ release $NO_2$ back into the atmosphere while wet/dry

deposition and aerosol uptake permanently remove $NO_x$ from the atmosphere. In TS2 aerosol uptake is the dominant fate of the organic nitrates consistent with previous studies (Fisher et al., 2016; Romer et al., 2016; Wolfe et al., 2015; Muller et al., 2018; Bates and Jacob, 2019) (Figure 12). Overall, TS2 has a greater proportion of organic nitrates undergoing aerosol uptake than TS1, which is largely driven by an increase in aerosol uptake of isoprene derived organic nitrates (Figure 12 and S7). There are also differences in the fraction of organic nitrates depositing. TS2 has more terpene derived organic nitrates depositing and

less isoprene derived organic nitrates depositing than TS1 (Figure 12 and S7).





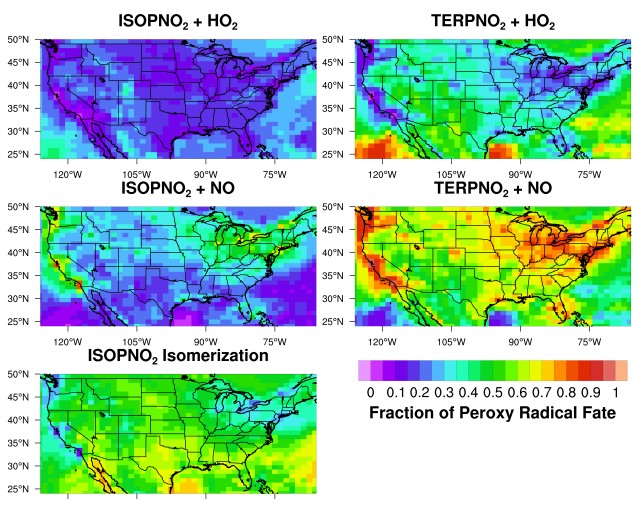

**Figure 11.** 2013 August average peroxy radical fate (fraction) below 2 km using TS2 for isoprene dihydroxy nitrate peroxy radical ($ISOPNO_2$) formed from isoprene 1st-generation hydroxy nitrate + OH (left) and terpene dihydroxy nitrate peroxy radical ($TERPNO_2$) produced from terpene unsaturated hydroxy nitrate + OH (right).

## 4.6 Chemical Mechanism Uncertainties Impact on Surface Ozone

Although our understanding of isoprene and terpene chemistry has substantially improved over the last decade, many uncertainties remain generally from disagreement between measurements for 1st-generation chemistry and a lack of experimental data for later-generation chemistry. A number of sensitivity tests were performed in order to evaluate how these uncertainties

impact simulated surface ozone. Each sensitivity test is briefly described in Table 1 and more completely described in Table S8. We note that the uncertainties evaluated here are only the known uncertainties and we prioritize sensitivity tests related to organic nitrate formation and fate. The bias in simulated surface ozone is still large in CESM/CAM-chem (Figure 7), so contributions from unknown chemistry are also possible.

### 4.6.1 Uncertainties in Formation of Organic Nitrates

The measured 1st-generation isoprene nitrate yield has been converging in recent experimental studies (Jenkin et al., 2015), but a moderately sized uncertainty still remains (0.09-0.13) (Xiong et al., 2015; Teng et al., 2017). This study uses $\sim 0.13$ consistent with Teng et al. (2017). The CIMS technique in Teng et al. (2017) is potentially more reliable for measuring isoprene hydroxy nitrates than that used by Xiong et al. (2015) because the CIMS in Teng et al. (2017) has nearly equal sensitivities for all isoprene hydroxy nitrate isomers while the CIMS in Xiong et al. (2015) varies in sensitivity depending on the isomer.

As shown in Figure 13a, this uncertainty is still important. Using the 0.09 yield from Xiong et al. (2015) increases the MDA8 surface ozone up to 2.6 ppb. Consistent with other systems, the 1st-generation isoprene chemistry is significantly better resolved than the 2nd-generation. In TS2 the later-generation organic nitrate yields are largely based on a parameterization developed





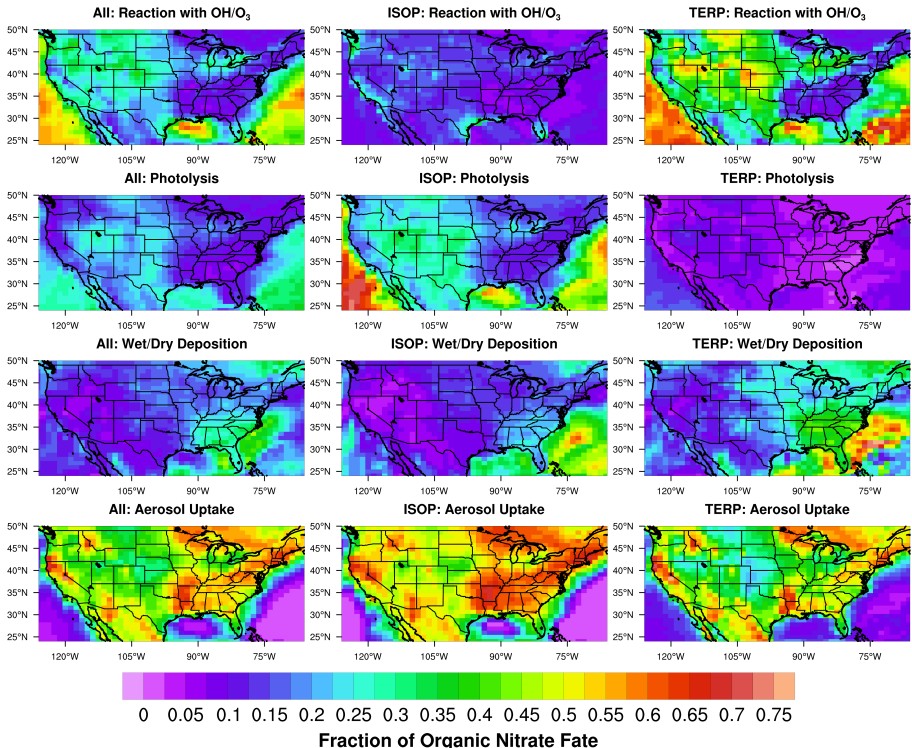

**Figure 12.** 2013 August average organic nitrate fate below 2 km using TS2 for all organic nitrates (left), isoprene organic nitrates (middle), and terpene organic nitrates (right). To avoid double counting, only the final fate is included, so reaction with OH/O$_3$ to form another organic nitrate is omitted from this calculation.

by Wennberg et al. (2018). If the 2$^{nd}$-generation nitrate yield is increased to an upper level limit of 0.3, modest changes in the MDA8 surface ozone occur with a decrease of up to 2.3 ppb (Figure 13b). When isoprene 1$^{st}$-generation hydroxy nitrates react with OH, the peroxy radicals that form possibly undergo isomerization reactions (Wennberg et al., 2018), but this pathway is quite uncertain and not well studied. Removing this isomerization pathway also has modest impacts on MDA8 ozone (Figure 13c).

For terpenes, even the 1$^{st}$-generation nitrate yields are not well constrained (Section 2.3.1). For $\alpha$-pinene, $\beta$-pinene, and limonene generally past experimental studies found that the organic nitrate yields varied from 0.15-0.26 (Noziere et al., 1999; Ruppert et al., 1999; Capouet et al., 2004; Rindelaub et al., 2015; Ruppert et al., 1999) with one study measuring a very low organic nitrate yield (0.01) from $\alpha$-pinene (Aschmann et al., 2002b). A more recent study (Xu et al., 2019) measured substantially lower hydroxy nitrate yields from $\alpha$-pinene (0.033) and $\beta$-pinene (0.064) with slightly higher total organic nitrate yields - $\alpha$-pinene (0.09) and $\beta$-pinene (0.11). As Xu et al. (2019) acknowledge, their instrument is not sensitive to organic nitrates from the H-abstraction pathway bringing their totals close but not quite to the lower end of the previous studies. Understanding why low organic nitrate yields from $\alpha$ and $\beta$-pinene have been measured in some cases is quite important.





**Table 1.** Sensitivity tests to determine the impact of uncertainties in TS2 on simulated surface ozone.

| Name | Description |
|------|-------------|
| **Uncertainties in Isoprene Chemistry** | |
| I_TEST1 | Isoprene 1st-gen organic nitrate yield updated to 0.09 from ∼ 0.13 |
| I_TEST2 | Isoprene 2nd-gen organic nitrate yield updated to 0.3 from 0.02-0.2 |
| I_TEST3 | Isoprene 2nd-gen no isomerization |
| **Uncertainties in Terpene Chemistry** | |
| T_TEST1 | Terpene 1st-gen organic nitrate yield updated to 0.3 |
| T_TEST2 | Terpene 1st-gen organic nitrate yield updated to 0.15 |
| T_TEST3 | APIN and BPIN 1st-generation chemistry based on Xu et al. (2019) |
| T_TEST4 | Terpene 2nd-gen MCM v3.3.1 organic nitrate yields for pinonaldehyde and limonaldehyde/limaketone oxidation. |
| **Uncertainties in Aerosol Uptake of Organic Nitrates** | |
| A_TEST1 | Aerosol uptake of all organic nitrates turned off |
| A_TEST2 | $\gamma$-values similar to GEOS-chem (Fisher et al., 2016) |
| A_TEST3 | $\gamma$-values similar to Wolfe et al. (2015) |

Additionally, greater understanding of how structure impacts the organic nitrate yield is especially valuable for terpenes given their large variety of chemical structures and reactivities (Guenther et al., 2012).

As shown in Figure 13d and e, increasing the 1st-generation organic nitrate yield to 0.3 or decreasing the organic nitrate yield to 0.15 for all terpenes impacts simulated MDA8 surface ozone by up to a couple ppb in either direction. Decreasing the 1st-generation organic nitrate yield from terpenes to 0.15 has a similar impact on simulated surface ozone as decreasing the isoprene 1st-generation organic nitrate to 0.09. Using the assumptions from Xu et al. (2019) (Figure 13f), also has modest impacts on simulated surface ozone increasing the MDA8 by up to 1.8 ppb. The changes from Xu et al. (2019) are likely moderate because only $\alpha$ and $\beta$-pinene oxidation was updated and although the organic nitrate yield is lower, the chemistry is quite similar to the theoretical studies that were used to develop TS2 (Vereecken et al., 2007; Vereecken and Peeters, 2012). Uncertainties in the later-generation organic nitrate yields has a large impact on ozone. When nitrate yields from MCM were used for pinonaldehyde and limonaldehyde oxidation rather than the parameterization in Wennberg et al. (2018) used throughout TS2, MDA8 ozone increased by up to 2.8 ppb (Figure 13g).



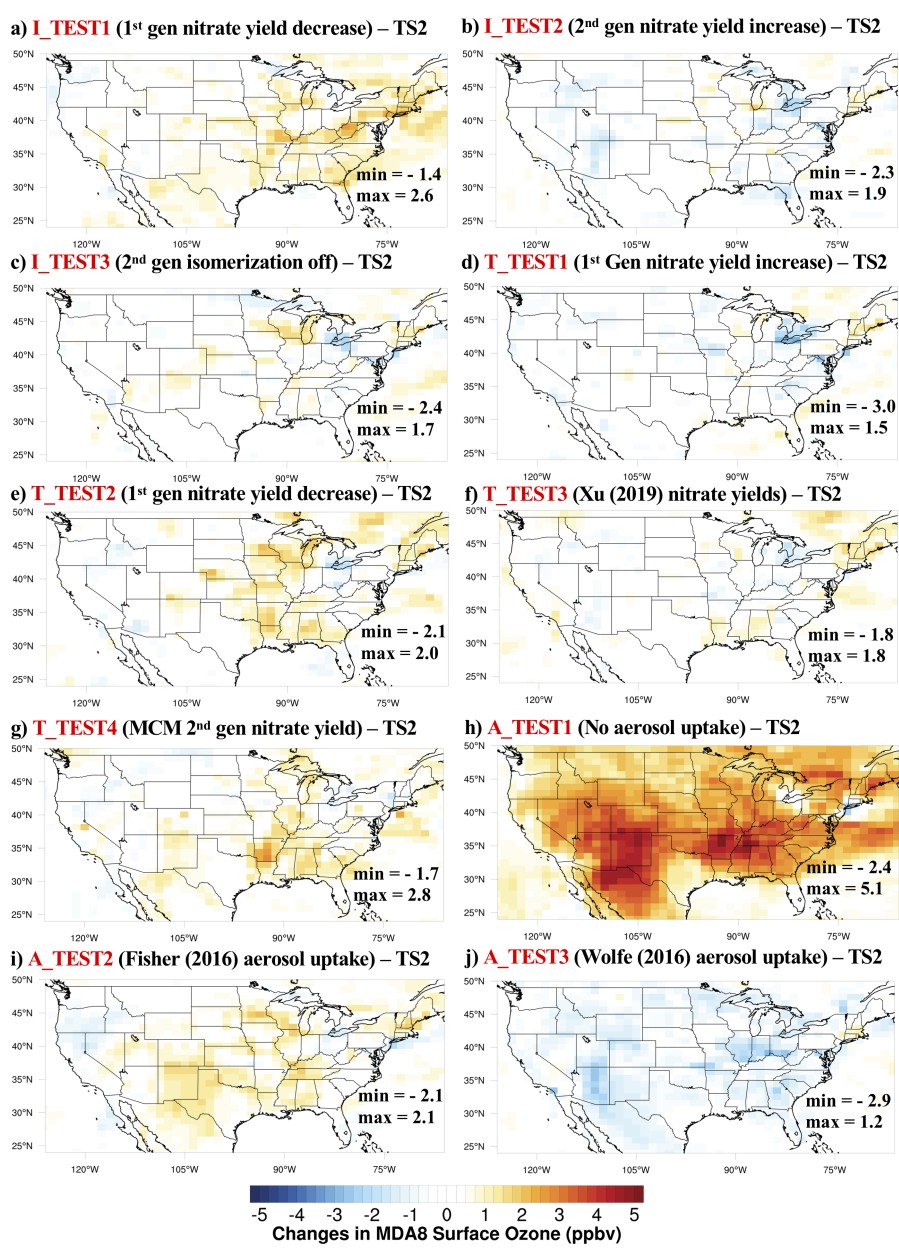

**Figure 13.** Difference in surface ozone daily max 8-hr average (MDA8) averaged over August 2013 between each sensitivity test described in Table 1 and the TS2 case. For each case, the minimum and maximum over the domain pictured is displayed on the bottom right.

In general, the uncertainties in isoprene and terpene oxidation evaluated here lead to similar impacts on simulated surface ozone even though terpene emissions are lower than isoprene in the southeast U.S. Uncertainties in isoprene chemistry have been significantly reduced over the last decade (Wennberg et al., 2018), such that uncertainties in terpene chemistry contribute





equally to uncertainties in ozone. These results suggest more experimental constraints on terpene chemistry would be beneficial to further reduce biases in simulated surface ozone. Based on the results from these terpene sensitivity tests, there is a clear need for more organic nitrate yield measurements from a variety of terpenes and terpene oxidation products with different chemical structures.

### 4.6.2 Uncertainties in Loss of Organic Nitrates

The uncertainties in aerosol uptake of organic nitrates causes the largest impact on simulated MDA8 (Figures 13h-j). Aerosol uptake was only recently identified as an important loss process for organic nitrates (Fisher et al., 2016; Romer et al., 2016; Li et al., 2018; Muller et al., 2018; Bates and Jacob, 2019). If no aerosol uptake is included in TS2, this increases the MDA8 ozone by up to 5.1 ppb (Figure 13h). Interestingly, in contrast Li et al. (2018) concluded aerosol uptake had little impact on
simulated ozone using GFDL AM3, which is likely caused by differences in the representation of the organic nitrate fate. For example, in Li et al. (2018) when aerosol uptake is removed, the $\beta$-isoprene hydroxy nitrate surrogate will react with OH to dominantly form other organic nitrates, so there is little expected impact on ozone. Conversely, in TS2 when aerosol uptake is removed, ISOPN2B, the $\beta$-isoprene hydroxy nitrate isomer that undergoes aerosol uptake (Figure 1), will instead react with OH to predominantly release $NO_2$ (Wennberg et al., 2018) and ozone will increase.

Most models like CESM/CAM-chem represent aerosol uptake of organic nitrates fairly simply (Section 2.4). Past studies indicate that aerosol uptake of organic nitrates is highly uncertain and under-constrained leading to quite a few different model implementations. For example, Fisher et al. (2016) determined bulk aerosol uptake coefficients for all isoprene ($\gamma = 0.005$) and terpene ($\gamma = 0.01$) hydroxy nitrates were needed when comparing GEOS-Chem model results with SEAC[4]Rs observations. Wolfe et al. (2015) used SEAC[4]Rs observations over the Ozark Mountains to estimate an aerosol uptake coefficient for isoprene
hydroxy nitrates ($\gamma = 0.02$). More recently, Bates and Jacob (2019) increased the tertiary organic nitrate aerosol uptake rate in GEOS-Chem by a factor of 10 compared to previous versions and Muller et al. (2018) used an aerosol uptake coefficient of 0.1 for all isoprene tertiary organic nitrates and 0.01 for all terpene organic nitrates.

From numerous laboratory studies (Jacobs et al., 2014; Hu et al., 2011; Darer et al., 2011), it is now clear that tertiary organic nitrates will undergo aerosol uptake and rapid hydrolysis while largely primary and secondary organic nitrates will not. Thus,
in TS2 all tertiary organic nitrates undergo aerosol uptake with the aerosol uptake coefficient measured by Wolfe et al. (2015), and largely primary and secondary organic nitrates are lost by other processes such as oxidation, photolysis, or deposition. To evaluate the range of possibilities, when all isoprene and terpene derived organic nitrates (tertiary, primary, and secondary) underwent aerosol uptake, but with the lower aerosol uptake used by Fisher et al. (2016), the MDA8 ozone increased by up to 2.1 ppb (Figure 13i). On the other hand, when aerosol uptake of all isoprene and terpene derived organic nitrates occurred with
the higher aerosol uptake from Wolfe et al. (2015), the MDA8 ozone decreased by up to 3 ppb (Figure 13j).

Considering the large impact aerosol uptake of organic nitrates has on simulated surface ozone (Figure 13) and the varied interpretations of past model studies explained above, more constraints are clearly needed. There have been noticeably more studies evaluating the formation of organic nitrates than their losses, but uncertainties in loss processes like aerosol uptake have substantial impacts on simulated ozone (Figure 13), and thus deserve equal attention.



## 5   Conclusions

The overall objective of this study is to add more observationally-based constraints to models (e.g., results from laboratory kinetic and product studies), in order to more realistically describe the chemistry and reduce uncertainty introduced by lumped species. To do this, more complex and updated isoprene and terpene chemistry was added into the default MOZART-TS1

mechanism in CESM2/CAM-Chem creating a new mechanism called MOZART-TS2. TS2 has isoprene chemistry with similar complexity as other recent work (Bates and Jacob, 2019; Muller et al., 2018) and terpene chemistry significantly more complex than past work (Browne et al., 2014; Fisher et al., 2016). The TS2 scheme could be easily adapted for use by other models. Although the focus of this work was to evaluate the impact of biogenic VOCs on simulated ozone, the TS2 mechanism will also be useful for other emission sources. For example, the use of volatile chemical products (McDonald et al., 2018a) and also

biomass burning (Koss et al., 2018) lead to the emission of a variety of terpenes into the atmosphere. TS2 with more speciated terpenes and more complex chemistry will be useful for accurately simulating ozone from these systems as well.

The box modeling results (Section 4.1 and 4.2) demonstrate that TS2 simulates 1st- and later-generation isoprene and terpene oxidation products better than TS1 in comparison to more explicit schemes like the Caltech mechanism and MCM, which enhances confidence that ozone is accurately simulated for the right reasons. Comparisons between TS2 and MCM terpene

chemistry demonstrate that MCM is inaccurately representing the oxidation chemistry for terpene unsaturated hydroxy nitrates leading to large impacts on $NO_x$ recycling and ozone formation.

Global model simulations using CESM/CAM-chem demonstrate enhanced capability of TS2 versus TS1 in simulating MDA8 surface ozone compared to CASTNET monitoring data with a reduction of biases up to 7 ppb in the eastern U.S. (Section 4.3). TS2 more accurately represents ozone, ozone precursors, and $NO_x$ reservoir species than TS1, when compared

to the SEAC[4]RS field campaign data (Section 4.4). Considering TS2 greatly increases the accuracy of isoprene and terpene chemistry, the $\sim 50\%$ increase in computational cost caused by adding 86 new species is reasonable. In particular, results from this study demonstrate the importance of terpene chemistry, which has been heavily reduced or ignored in models in the past. Future work is needed to improve terpene schemes used in other models, to measure more terpene oxidation products in the field, and to conduct more laboratory studies to better understand terpene oxidation.

Comparisons of TS2 with the SEAC[4]RS field campaign data also demonstrate remaining biases. Although substantial changes were made to organic nitrate formation and fate in TS2 compared to TS1 (Section 4.5), there remain large biases in modeled and observed total organic nitrates, which are not unique to CESM/CAM-chem (e.g., Fisher et al., 2016). If these missing organic nitrates have large losses, they could help to explain the remaining bias in simulated surface ozone.

Although our understanding of isoprene and terpene chemistry has increased significantly over the last several decades,

many lingering uncertainties in the chemistry remain. A number of sensitivity tests were performed in order to evaluate how uncertainties in isoprene and terpene chemistry with a focus on organic nitrate formation and fate translate to uncertainties in simulated MDA8 ozone (Section 4.6). Even though isoprene emissions are much larger than terpene emissions, because isoprene chemistry is better understood, uncertainties in isoprene and terpene chemistry have similarly modest impacts on MDA8 surface ozone. An increased understanding of terpene oxidation including later-generation chemistry will provide important



further constraints to CESM/CAM-chem. Considering the diversity of terpenes and their oxidation products, measurements of organic nitrate yields from all compounds is not practical. Perhaps, measurements from a small subset of terpene and terpene oxidation products with wide structural differences may be particularly useful for identifying patterns and creating parameterizations that will reasonably extrapolate to the rest.

Past work has largely focused on formation processes of organic nitrates, but the sensitivity tests performed here (Section 4.6) suggest loss processes are equally important and quite under-constrained. In particular, future work to better understand aerosol uptake of organic nitrates and to incorporate more complex parameterizations in models is needed. In the future organic nitrate aerosol uptake will be updated to a more complex scheme in CESM/CAM-chem. Organic nitrates will undergo aerosol uptake based on their volatility like the volatility basis set scheme for SOA formation combined with the addition of hydrolysis

reactions in the particle-phase to form nitric acid. Hydrolysis rates of many organic nitrates have already been measured in the laboratory (e.g., Liu et al., 2012; Jacobs et al., 2014; Rindelaub et al., 2015) or inferred from field campaign observations (Lee et al., 2016; Romer et al., 2016). Additional processes to represent aqueous-phase chemistry of organic nitrates loss in clouds through a similar process as that which occurs on aerosols should also be considered and included. Currently in CAM-chem, organic nitrates are lost to clouds via wet deposition, but do not undergo hydrolysis in the cloud to form $HNO_3$. Because tertiary

nitrates have very different loss processes than secondary/primary nitrates, a greater understanding of the isomer distribution is also needed. Recent studies investigating the isomer distribution of various compounds have been conducted using both experimental (Teng et al., 2017; Boyd et al., 2015; Xu et al., 2019) and theoretical (Vereecken et al., 2007; Vereecken and Peeters, 2012) approaches, but more future studies are needed to ensure the fate of organic nitrates is accurately represented in models.

Additionally, TS2 has been designed to include surrogates for isoprene and terpene SOA precursors, which is the beginning framework for coupling SOA formation directly to gas-phase chemistry in CESM/CAM-chem as recent studies have done with other models (Marais et al., 2016; Bates and Jacob, 2019; Stadtler et al., 2018; Zare et al., 2019). Considering the sensitivity of simulated ozone on aerosol uptake of organic nitrates (Figure 13), accurately representing SOA and ozone seems to be a more connected problem than previously recognized. Better coupling SOA formation to gas-phase chemistry in models is likely

important for accurately simulating both ozone and SOA.

As shown in Figure 7, TS2 greatly reduces the surface ozone bias in CESM/CAM-chem, but a relatively large bias remains. This bias could be caused by remaining uncertainties in the chemistry either by those described in Section 4.6 or by unknown uncertainties. Other factors, which will be evaluated in future studies, are likely also important including horizontal and vertical resolution, clouds, vertical transport in and above the planetary boundary layer, ozone deposition, anthropogenic emissions,

and biogenic emissions.

*Code and data availability.* CESM2.1.0 is a publicly released version of the Community Earth System Model available from http://www.cesm.ucar.edu/. The code updates described in this work will be made available as a new compset in future releases of CESM. If this is not possible, the code updates will be provided on the CAM-chem wiki at https://wiki.ucar.edu/display/camchem/Gas-Phase+Chemistry.



Please contact Rebecca Schwantes (rschwant@ucar.edu) if you would like these updates prior to their future release in CESM or need any of the CESM output used in this work. BOXMOX is also publicly available (https://boxmodeling.meteo.physik.uni-muenchen.de/). The MCM v3.3.1 mechanism, which was added into BOXMOX for this work, will be added to a future release of BOXMOX. Please contact Rebecca Schwantes (rschwant@ucar.edu) if you would like the BOXMOX mechanisms, initialization files, or output used in this work.

*Author contributions.*   RHS designed the study. RHS updated the isoprene and terpene chemistry with assistance from JJO, GST, and LKE. MCB determined all updates to Henry's Law constants for the MOZART-TS1 mechanism. RHS expanded this for the MOZART-TS2 mechanism and incorporated all values into CESM/CAM-chem. RHS performed all model simulations with help from LKE. SRH and KU collected actinic flux spectroradiometer measurements and calculated photolysis rate constants during the SEAC[4]RS field campaign. JMSC measured isoprene oxidation products with the Caltech $CF_3O^-$ CIMS. DRB measured carbon monoxide, isoprene, and $\alpha$- & $\beta$-pinene using the whole

air sampler during SEAC[4]RS. AW measured acetonitrile and monoterpenes using proton transfer reaction - mass spectrometry (PTR-MS) during SEAC[4]RS. TVB measured wind and temperature during SEAC[4]RS.

*Competing interests.*   The authors declare that they have no conflict of interest.

*Acknowledgements.*   This material is based upon work supported by the National Center for Atmospheric Research, which is a major facility sponsored by the National Science Foundation (NSF) under Cooperative Agreement No. 1852977. The CESM project is supported primarily

by the NSF. Computing and data storage resources, including the Cheyenne supercomputer (doi:10.5065/D6RX99HX), were provided by the Computational and Information Systems Laboratory (CISL) at NCAR (CISL, 2017). We thank Simone Tilmes for assistance with CESM simulations. We thank the following for collecting SEAC[4]RS field campaign data: Paul O. Wennberg and John D. Crounse for measuring isoprene oxidation products with the Caltech $CF_3O^-$ CIMS; Ronald C. Cohen for measuring $NO_2$, organic nitrates, and total peroxy acyl nitrates with the thermal dissociation-lacer induced fluorescence (TD-LIF) instrument; and Thomas B. Ryerson and the NOAA NOyO3 team

for measuring NO, $NO_2$, and $O_3$. PTR-MS VOC measurements aboard the NASA DC-8 during SEAC[4]RS were supported by the Austrian Federal Ministry for Transport, Innovation and Technology (bmvit) through the Austrian Space Applications Programme (ASAP) of the Austrian Research Promotion Agency (FFG). Tomas Mikoviny is acknowledged for his support during SEAC[4]RS with the PTR-MS. We thank Alma Hodzic and Siyuan Wang for helpful discussions.



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
