# Peer review of "Comprehensive isoprene and terpene gas-phase chemistry improves simulated surface ozone in the southeastern U.S."

_Atmospheric Chemistry and Physics, 2019_

## Referee Comment (RC1) · Anonymous Referee #1 · 14 Nov 2019

In the manuscript, the authors have updated the isoprene and terpene chemistry in CESM/CAM-chem model to determine its impact on simulations of ozone. With the updates predominantly focused on organic nitrate production and fate, the bias in mean ozone concentrations was reduced by up to 7 ppb. The comprehensive study involved box model simulations and field study comparisons where the CESM/CAM-chem model was compared with more explicit atmospheric chemistry models such as MCM and the Caltech isoprene mechanism. While mean biases were reduced in the revised model, it still does not capture the vertical profile of ozone suggesting other chemical and/or physical processes are responsible which must be discussed further. The manuscript would have to be revised in order to be accepted for publication in ACP.

[Figure]

Major comment:

As the authors indicate on multiple occasions, it is imperative to model atmospheric ozone better for the right reasons. This is a high standard to achieve in atmospheric modeling because numerous processes (both chemical and physical) may affect the concentration of a singular important species in the atmosphere. The first justification that the updated chemistry in TS2 improved the simulations of ozone for the right reasons was that its output in box modeling simulations closely matched that of more explicit mechanisms like the MCM as shown in Figures 3-6. This was summarized in the second paragraph of the Conclusions. However, in that same paragraph the authors highlighted important differences between TS2 and MCM seemingly contradicting their previous statement. Inter-model comparisons are not extremely useful in justifying updates to a model because both models may contain quite a few simplifications and/or uncertainties. A better justification was made when comparing the model outputs with experimental data. While Figure 7 illustrates that TS2 does reduce the bias of modeled surface ozone concentrations across the US when compared with US EPA CASTNET data, Figure 9 (1st panel) shows that the model vertical profile is still inaccurate with overpredictions below the planetary boundary layer and underpredictions above it when compared with SEAC4Rs flight tracks. While the changes involving predominantly the organic nitrate chemistry have reduced surface level biases in ozone predictions, challenges remain including: "PBL height, mixing schemes, clouds, vertical resolution, or ozone dry deposition schemes" (bottom of pg. 25). This should be echoed in the abstract and conclusions with further discussion in the section from which the quote was extracted (Section 4.4 after pg. 25). While the discussion of Figure 9 is extensive, model biases for different observables should be tied back to the ozone profile since this is the focus of the paper. For example, can corrections to jNO2 or the concentrations of biogenic VOCs be enough to predict the correct shape of the ozone vertical profile?

Minor comments:

1. Reference to CMAQ isoprene updates should be added to the Introduction on page 2 line 29 (see Pye et al. Environ. Sci. Technol. 2013, 47, 19, 11056-11064)

2. It was unusual not to see some of the organic nitrates in Figures 1 and 2 in blue when they are isomers of other species that undergo aerosol uptake. It isn't until 2.2.1 (pg. 7 line 7) that the author's mention tertiary nitrates are more likely to experience reactive aerosol uptake. This could also be specified in the captions to avoid confusion.

3. Are there gray boxes that can be added to Figure 2 like in Figure 1 to denote new chemistry?

4. dH/R should precede "6014 K" on line 16, page 5 of the manuscript for clarity of where the value comes from.

5. The yield of IEPOX from ISOPOOH + OH should be stated in the text on page 6 in order to understand its contribution to recycling HOx and aerosol formation. Other products of ISOPOOH + OH include the formation of isoprene dihydroperoxides (ISOPOOHOOH) that consume HOx (and therefore may give rise to less ozone) as in Liu et al. Environ. Sci. Technol. 2016, 50, 9872-9880 and Piletic et al. J. Phys. Chem. A, 2019, 123, 4, 906-919. Why were such products that potentially affect HOx, ozone and aerosol yields not included in an updated isoprene mechanism?

6. Many papers regarding Criegee intermediates have the 'C' capitalized. Please correct this in the manuscript.

7. On page 17, line 8, it states that "TS2 was altered to use RCIM assumptions for formation and loss of PAN and..." What specifically was altered in this sensitivity test?

8. On page 19, line 18, it states "Commonly, in MCM, unsaturated hydroxy nitrates will react with OH via H-abstraction ...". Please add "derived from terpenes" after nitrates to specify that this does not involve hydroxy nitrates derived from isoprene.

9. Figure 9 should have same color scheme for TS1 and TS2 with its predecessors (Figs 3-6) for consistency.

10. The equation used to derive the organic nitrate yields using alpha and n in the Supporting Information (pgs. 49-51) should be stated with the reference included for clarity.

---

## Referee Comment (RC2) · Anonymous Referee #2 · 15 Nov 2019

Anonymous Review of Schwantes et al. 2019

This is a substantial manuscript describing the development and testing of a new comprehensive gas-phase isoprene and terpene chemical mechanism (TS2) within the CESM/CAM-chem model as an update to the previous chemical mechanism TS1. TS2 adds an additional 25 terpene and 21 isoprene transported species with 219 terpene and 139 isoprene reactions which, when included in the CESM/CAM-chem model increase run times by ~ 50%. The manuscript details the development process (Section 2), the modeling used to test the mechanism (both box modeling and global modeling) and comparisons to more explicit mechanisms (MVM, RCIM) for the box modeling and comparisons to observational datasets (CASTNET and SEAC4RS) for the global modeling. To characterize uncertainties and test sensitivities the authors perform a suite of sensitivity tests, and the authors present recommendations for future directions of study and suggest future work to further constrain remaining uncertainties. The supplemental material is substantial and comprehensive, and includes sufficient detail to enable future work and development of the TS2 mechanism. The mechanism is expected to be included in future versions of the CESM model and is offered to any interested party if they'd like to access the updates prior to the new release. If for whatever reason the TS2 mechanism does not make it into a future CESM release the authors should guarantee an alternative location for the code to be made available.

I feel that this manuscript is an excellent one and does not need any major revisions to be accepted and published in ACP. Both the significance and quality of the science is high. Although I do think a slight restructuring of Section 4 would aid in clarity (described below). Here I will move through the major sections and offer (mostly minor) suggestions.

The Introduction (Section 1) is well-written and I have no suggestions.

The description of the development of TS2 (Section 2) is long (nearly 10 pages) and very detailed. It is also very well-organized and is supported by supplemental Figures S1 and S2 and supplemental Tables S1, S2, S3, S4, S5, and S6. Figures 1 and 2 should point to Table S2. This section summarizes (in detail) updates to the Henry's Law Coefficients and updates to isoprene and terpene chemistry, as well as (only briefly) the aerosol scheme. A few minor suggestions:

- Figures 1 and 2 are schematics of OH oxidation and Figures S1 and S2 are schematics of $NO_3$ oxidation. Why were no schematics of $O_3$ oxidation included (and only described in Sections 2.2.2 and 2.3.2). For completeness, I feel that $O_3$ oxidation schematics could be included in the supplement.
- Page 7, Line 1: The authors mention that "multiple isomers were only incorporated into TS2 if grouping them together would bias $HO_x$ or $NO_x$ budgets." How was this determined? Were there sensitivity or testing runs conducted and that are not described? Please describe.
- Page 7, Line 9: The authors mention that the "number of organic nitrates in TS2 was optimized to accurately represent" varying reaction rates. What sort of optimization was done? What were the optimization criteria? This needs more detail.

- The supplemental Figures S1 and S2 should be referenced more thoroughly in the manuscript, esp. within the captions of Figures 1 and 2 and in the accompanying sections (Section 2.2.3 and 2.3.3).
- Page 10, Line 10: Were there any alternative approaches that could be made to represent terpene products here? Were these approaches tested or considered? A description of why this particular choice was made (and what the likely impact on computational cost and resulting chemistry would be) should be included.
- Section 2.4: I feel that a more thorough descriptions of the impacts of the TS2 mechanism on aerosol formation can be included. The existing section is sparse compared to the other sections, and although the authors make not of the large uncertainties that remain, I feel that more can be added. Perhaps the manuscript title can include the word "gas-phase" after the word "Comprehensive" in order to highlight that this manuscript is largely focused on the gas-phase chemistry of isoprene and terpene and that future development of the aerosol scheme is slated for future work.

The methodology section (Section 3) is well written and comprehensive and I have no suggestions.

The results and discussion section (Section 4) is very long (21 pages) and I feel can be split into separate sections. Perhaps the evaluation of the explicit schemes (Sections 4.1 and 4.2) and the global modeling evaluation against surface observations and field campaign data (Sections 4.3 and 4.4) can remain in Section 4, while the discussion of the organic nitrate fate (Section 4.5) and the overall discussion of uncertainties (Section 4.6) can be moved to a Discussion Section (Section 5). The existing Conclusion section (Section 5) can then me moved to Section 6.

The comparisons to box model (Section 4.1 and 4.2) is well-presented and well-supported. The dance between the figures and tables in the manuscript and the supplement at times makes it difficult to "follow the thread," but I believe the authors have done a good job in allowing a reader to either stick to the manuscript or follow the details provided in the supplemental material. A few minor comments:
- In Figures 3, 4, 5, and 6, I find it somewhat difficult to differentiate between the blue colors used to label TS1 and TS2. Changing one of them to a more distinct color (green, perhaps?) would aid in clarity. [NOTE: This appears to be the case only when I print the manuscript out, and the colors are clearly distinguishable when I look at a digital copy]
- The sensitivity tests in the Figures are labelled as "Test 1, Test 2", while in the captions are labelled "test 1, test 2" and in the manuscript as "first sensitivity test, second sensitivity test." I suggest sticking with the "Test 1, Test 2" format and making this consistent throughout the sections.
- Page 19, Line 26: The authors state that "These unrealistically fast cycles are commonly used in MCM to reduce the number of species...". Is there a citation for this claim? How do the authors know this is the case?

The comparison to surface and field campaign data (Section 4.3 and 4.4) is well presented and I have no comments.

The presentation of organic nitrate formation and fate (Section 4.5) and the discussion pertaining to mechanism uncertainties (Section 4.6) and the Conclusion (Section 5) is also well presented. I have a few minor comments:

- Page 31 Line 7-8: Could you expand (speculate) on some of the possible sources of uncertainties that are less-well known?
- Page 34, Line 2 – Page 35, Line 1: This point has also been made by Mao et al. (2018) (www.atmos-chem-phys.net/18/2615/2018/) (see page 2622), and should be referenced.
- Page 37, Line 28: I suggest including the Sun et al. (2017) reference in which they were able to reduce Eastern US ozone bias via the utilization of a new solver scheme (https://agupubs.onlinelibrary.wiley.com/doi/full/10.1002/2016MS000863). Additionally, I believe a slightly expanded discussion of the remaining sources of uncertainty would significantly enhance this final paragraph.

---

## Referee Comment (RC3) · Anonymous Referee #3 · 21 Nov 2019

Review of "Comprehensive isoprene and terpene chemistry improves simulated surface ozone in the southeastern U.S." by R.H. Schwantes et al. (manuscript #acp-2019-902).

This manuscript describes the development of a detailed scheme for photooxidation of isoprene and terpenes in CESM/CAM-chem. The new scheme (MOZART-TS2) includes a significantly more complex representation of isoprene and terpene chemistry compared with the default mechanism (MOZART-TS1), with 21 additional transported species and over 100 additional reactions. The chemistry updates are informed by observationally based constraints from recent experimental data and explicit mechanisms

(MCMv3.3.1 and the Caltech isoprene mechanism). While still greatly reduced compared with explicit mechanisms, MOZART-TS2 includes significantly more detail than other mechanisms used in chemistry-climate models, particularly in the case of terpene chemistry. The new scheme is tested against these two explicit scheme in a box model context, and its broader impacts are assessed against SEAC4RS and USEPA CASTNET observations in the context of CESM/CAM-chem.

The study finds, in a box model context, that TS2 improves the simulation isoprene and terpene oxidation products over TS1, in comparison with explicit schemes. Several discrepancies between TS2 and MCM are attributed to faulty assumptions included in MCM. In the global model context, TS2 is found to reduce surface ozone biases and, in general, more accurately represent ozone, ozone precursors, and NOx reservoir species than TS1, in comparison with field observations. The new chemistry scheme, however, imposes a significant cost, increased in the computational time of CESM/CAM-Chem by ∼50%.

This manuscript is quite comprehensive and generally well written. The mechanisms, while complex, are presented clearly. Sufficient motivation and justification are provided for the decisions made. The evaluation, using both box models and global chemistry-climate models, give a very clear sense of the impacts of this mechanism on tropospheric chemistry, in particular on ozone. Some of the significant uncertainties remaining in the chemistry are explored and documented through carefully selected sensitivity simulations. While the manuscript is quite lengthy, this seems appropriate given the high level of detail being presented. Suitable use is made of supplementary material to convey somewhat lower-priority information. Overall, this is a very impressive piece of work. With relatively modest revisions, as described below, this paper would be suitable for publication in ACP, and would provide a valuable reference for chemical mechanisms of BVOC oxidation.

General comments

1) The box model simulations, comparing TS2 with TS1 and explicit mechanisms, currently only uses one set of chemical and meteorological conditions (August in Mississippi). It would have been helpful for box model simulations to have been conducted separately for high-NOx and low-NOx conditions. Given the very different oxidation pathways under these conditions, this would have provided a more stringent set of tests for the new mechanism. (I realize that new simulations would impose a significant analysis burden on the authors, so may not be practical.)

2) It might be helpful to split Results (4.1-4.4) and Discussions (4.5-4.6) into separate sections.

Specific comments

1. Introduction

page 3, lines 11-12 – Rephrase for clarity. For instance, "to determine the extent to which [improvements to] the chemical mechanism can explain ...."

2.1 Updates to Henry's Law Constants

p.5, l.15-16 – Clarify the definition of Henry's law temperature dependence (given here as 6014 K).

p.5, l. 16 – Add "used for dry deposition" after "reactivity factor (F0)."

3.1 Box modeling

p.13, l.24 – Planetary boundary layer *height*?

p.13, l.24 – Clarify here that only "general" photolysis rate constants are taken from CESM/CAM-chem-TS1 (as explained later).

p.13, l. 27-28 – Explain how deposition from CESM/CAM-chem is implemented in box model. For instance, dry deposition velocities? wet deposition loss frequencies? No ventilation/dilution of the box with background air is included, correct?

p.14, l.13-14 – "These are ideal scenarios designed" to "These idealized scenarios are designed ...."

3.2 Global modeling

p.14, l.18-21 – Which meteorological fields are nudged to reanalysis?

p.15, l.4-5 – Which years were used for spinup?

4. Results and Discussions

Figures 3-6 – Difficult to distinguish some of the individual lines in these plots. Try to modify colors, or make lines thicker.

p.17, l.8-9 – Explain the differences between the representations of PAN formation and loss (TS2 versus RCIM).

p.17, l.11 – Add "from RCIM" after "The PAN assumptions."

p.17, l.13 – Are the RCIM photolysis rates faster or slower? By how much?

4.2 Terpene Evaluation Against Explicit Species

p.19, l.5 – Clarify what is meant here by "total products produced."

p.20, l.3 – Add "oxidation of" before "the alpha-pinene."

4.4 Evaluation Against Field Campaign Data

Figure 7 – Add mean bias for Eastern US / Western US to figure panels.

p.28, l.1-3 – How do the dry deposition velocities of OVOCs compare in GEOS-Chem versus CESM/CAM-chem?

4.6.2 Uncertainties in Loss of Organic Nitrates

p.35, l.24 and l.26 – Clarify the meaning of "largely" here, e.g., do you mean "primary and secondary organic nitrates *largely* will not" and "... are *largely* lost ...."

Technical corrections

1. Introduction

p.3, l.3 – Hyphenate "terpene-derived" (and "isoprene-derived" throughout manuscript).

2. Development of MOZART-TS2

p.3, l.11 – Capitalize "Model."

3.1 Box modeling

p.14, l.6 – Include units for lat/lon (deg N, deg E).

3.2 Global modeling

p.14, l.22 – Run-on sentence. Break into two, starting with "Using a weak ...."

p.14, l.34 – Change to "(Table S3), using ...."

4. Results and Discussions

p.15, l.15 – "suggests" –> "suggest"

p.17, l.1 – Hyphenate "NO3-initiated."

4.2 Terpene Evaluation Against Explicit Species

p.20, l.1 – "Terpene-rich"

4.4 Evaluation Against Field Campaign Data

p.27, l.3 – Change to "above 2km; when clouds ...."

4.5 Organic Nitrate Formation and Fate

p.28, l.34 – "isoprene- and terpene-derived."

p.29, l.6 – Delete comma.

4.6.1 Uncertainties in Formation of Organic Nitrates

p.33, l.6 – Delete comma.

4.6.2 Uncertainties in Loss of Organic Nitrates

p.35, l.13 – Delete comma after "(Figure 1)."

p.35, l.16 – "under-constrained, leading to ...."

5. Conclusions

p.36, lines 15, 18, 31 – Missing commas.

p.37, line 9 – Missing comma.

---

## Author Response (AR1)

**Response to Review #1**

Thank you for the helpful comments and suggestions. We appreciate your time for reviewing our paper. We have addressed all of your comments as detailed below:

**Major comment:**

**As the authors indicate on multiple occasions, it is imperative to model atmospheric ozone better for the right reasons. This is a high standard to achieve in atmospheric modeling because numerous processes (both chemical and physical) may affect the concentration of a singular important species in the atmosphere. The first justification that the updated chemistry in TS2 improved the simulations of ozone for the right reasons was that its output in box modeling simulations closely matched that of more explicit mechanisms like the MCM as shown in Figures 3-6. This was summarized in the second paragraph of the Conclusions. However, in that same paragraph the authors highlighted important differences between TS2 and MCM seemingly contradicting their previous statement. Inter-model comparisons are not extremely useful in justifying updates to a model because both models may contain quite a few simplifications and/or uncertainties.**

We revise the text in the conclusions to explain in more detail why in general MCM does compare well with MOZART-TS2, but due to some inaccurate assumptions in MCM there are differences between the two mechanisms that impact ozone formation. Although we agree that field campaign comparisons are very useful for evaluating new chemical mechanisms, we also believe inter-comparisons between different chemical mechanisms are valuable. Without such comparisons, it would be difficult for future studies to accurately pick the best chemical mechanism to use to answer their specific science question. Also we want to highlight the inaccurate assumption in MCM here, so that future studies that use MCM for terpene oxidation update this chemistry.

"The box modeling results (Section 4.1 and 4.2) demonstrate that TS2 simulates 1$^{st}$- and later-generation isoprene and terpene oxidation products better than TS1 in comparison to more explicit schemes like the Caltech mechanism and MCM. Verification that the isoprene and terpene oxidation chemistry is correctly represented in TS2 enhances confidence that ozone is accurately simulated for the right reasons. The box-modeling results also uncovered a simplification in MCM where terpene unsaturated hydroxy nitrate oxidation is not accurately represented, leading to large impacts on $NO_x$ recycling and ozone formation (Section 4.2). Future studies using MCM v3.3.1 to simulate the oxidation of terpenes, especially those that contain more than one double bond, should update this simplification."

**A better justification was made when comparing the model outputs with experimental data. While Figure 7 illustrates that TS2 does reduce the bias of modeled surface ozone concentrations across the US when compared with US EPA CASTNET data, Figure 9 (1st panel) shows that the model vertical profile is still inaccurate with overpredictions below the planetary boundary layer and underpredictions above it when compared with SEAC4Rs flight tracks. While the changes involving predominantly the organic nitrate chemistry have reduced surface level biases in ozone predictions, challenges remain**

**including: "PBL height, mixing schemes, clouds, vertical resolution, or ozone dry deposition schemes" (bottom of pg. 25). This should be echoed in the abstract and conclusions with further discussion in the section from which the quote was extracted (Section 4.4 after pg. 25). While the discussion of Figure 9 is extensive, model biases for different observables should be tied back to the ozone profile since this is the focus of the paper. For example, can corrections to jNO2 or the concentrations of biogenic VOCs be enough to predict the correct shape of the ozone vertical profile?**

Good points. We revise the text to further highlight how biases in various products like jNO2 and biogenic VOCs will influence ozone:

"The vertical profile shape for ozone is quite different between the model and observations. There are also clear differences between the model and observations for $NO_2$ photolysis and ozone precursors like $NO_x$ and VOCs. Possible explanations for the remaining biases are further described below and will be explored more completely in future work."

"A cloud-induced reduction in $j_{NO2}$ below 1 km and an increase between 2-4 km would improve the ozone and $j_{NO2}$ profile shapes."

"Future work will evaluate whether using finer horizontal resolution (14 km) removes these biases in biogenic VOCs in CAM-chem. If isoprene and monoterpenes are still under-predicted, increasing these biogenic emissions will enhance the overall bias in ozone."

We also add the following to the conclusions to highlight what the next steps will be in improving the ozone bias in CAM-chem in the future:
"This bias could be caused by remaining uncertainties in the chemistry (Section 5.2) or by processes other than chemistry, which will be evaluated in future work. Considering the analysis against the SEAC[4]RS field campaign results, the first step toward reducing the remaining ozone bias will be to evaluate how finer horizontal resolution (14 km) impacts the results. Because biogenic and anthropogenic emissions in the southeast U.S. are spatially segregated, improvements in simulated ozone and biogenic VOCs are expected with finer horizontal resolution. Future work will also include evaluating different anthropogenic emission inventories and a more thorough investigation into whether biogenic emissions are accurately represented by MEGAN in CAM-chem. Additionally, cloud biases in CAM-chem will be investigated more in the future given their likelihood for improving the vertical profile shape of ozone, ozone precursors, and $j_{NO2}$. Considering that biases in the ozone profile shape are enhanced with stronger nudging to meteorological data (Figure S6), a more thorough analysis on the impact of nudging on CAM-chem dynamics and cloud parameterizations should be conducted. Future work will also evaluate whether enhanced vertical resolution is needed to improve PBL height and mixing schemes. Further evaluation of different chemical solvers (Sun et al., 2017) is also needed. Additionally, ozone dry deposition has a large impact on simulated surface ozone (val Martin et al., 2014; Clifton et al., 2019) and a thorough evaluation and update to the ozone dry deposition scheme used in CAM-chem should be performed. Ozone is a complicated pollutant to accurately simulate in models. This work demonstrates that updating isoprene and terpene gas-phase chemistry clearly improves simulated surface ozone in CAM-chem and that additional studies evaluating and updating other processes are needed to further reduce the ozone bias."

And we added in the abstract:

"Although the updates to isoprene and terpene chemistry greatly reduce the ozone bias in CAM-chem, a large bias remains. Evaluation against SEAC$^4$RS field campaign results suggests future improvements to horizontal resolution and cloud parameterizations in CAM-chem may be particularly important for further reducing this bias."

**Minor comments:**

**1. Reference to CMAQ isoprene updates should be added to the Introduction on page 2 line 29 (see Pye et al. Environ. Sci. Technol. 2013, 47, 19, 11056-11064)**

Yes, thanks for pointing this out. We have added this reference.

**2. It was unusual not to see some of the organic nitrates in Figures 1 and 2 in blue when they are isomers of other species that undergo aerosol uptake. It isn't until 2.2.1 (pg. 7 line 7) that the author's mention tertiary nitrates are more likely to experience reactive aerosol uptake. This could also be specified in the captions to avoid confusion.**

Yes, we have added:

"As shown, only certain isomers of organic nitrates undergo aerosol uptake as explained in Section 2.4.

**3. Are there gray boxes that can be added to Figure 2 like in Figure 1 to denote new chemistry?**

Yes, we have added gray boxes as done in Figure 1 for clarity.

**4. dH/R should precede "6014 K" on line 16, page 5 of the manuscript for clarity of where the value comes from.**

Yes, we update this sentence to:

"If the Henry's law temperature dependence **(dH/R)** was unavailable in the literature, 6014 K was assumed consistent with GECKO-A."

**5. The yield of IEPOX from ISOPOOH + OH should be stated in the text on page 6 in order to understand its contribution to recycling HOx and aerosol formation. Other products of ISOPOOH + OH include the formation of isoprene dihydroperoxides (ISOPOOHOOH) that consume HOx (and therefore may give rise to less ozone) as in Liu et al. Environ. Sci. Technol. 2016, 50, 9872-9880 and Piletic et al. J. Phys. Chem. A, 2019, 123, 4, 906-919. Why were such products that potentially affect HOx, ozone and aerosol yields not included in an updated isoprene mechanism?**

The MOZART-TS2 mechanism does not treat ISOPOOHOOH as a separate species in order to reduce cost because it is a minor product. However, ISOPOOHOOH is included in the surrogate compound ISOPHFP, which represents all isoprene highly functionalized hydroperoxides and

there is a yield of ISOPHFP from ISOPOOH + OH reaction. Considering the yield of ISOPOOHOOH and other low-volatility products formed from 1,5-H-shifts from ISOPOOH + OH reaction is uncertain including a lumped tracer ISOPHFP for these processes is reasonable. ISOPHFP and other low-volatility products do have a loss to aerosol in CAM-chem, so that $HO_x$ and $O_3$ are more accurately represented, but they do not form SOA directly. In CAM-chem, SOA is formed via a VBS scheme, which was not updated in this work. Future work will link products like ISOPHFP to form SOA directly in CAM-chem, but such updates are beyond the scope of this work.

We update the title to "Comprehensive isoprene and terpene **gas-phase** chemistry improves simulated surface ozone in the southeastern U.S." to emphasize the scope of this work is to update the gas-phase mechanism.

We recognize that the description did not include enough detail for the isoprene low-$NO_x$ chemistry and so have added the following to the MOZART-TS2 mechanism description in Section 2.2.1 and cited the papers above:

"OH addition to ISOPOOH forms a 0.85 yield of isoprene epoxydiol (IEPOX) & OH; a 0.07 yield of glycolaldehyde, hydroxyacetone, & OH; and a 0.08 yield of ISOPHFP, which is a surrogate compound for all isoprene highly functionalized hydroperoxides (Krechmer et al. 2015; Riva et al., 2016; St. Clair et al., 2016; Liu et al., 2016; Piletic et al., 2019). ISOPHFP undergoes aerosol uptake in TS2 to more accurately represent loss processes of $HO_x$, but like organic nitrates does not explicitly form SOA. SOA is formed by a volatility basis set (VBS) scheme in CAM-chem, which was not updated in this work. Directly forming SOA from low-volatility products like ISOPHFP in CAM-chem is a goal for future work."

**6. Many papers regarding Criegee intermediates have the 'C' capitalized. Please correct this in the manuscript.**

Thanks, we have updated this.

**7. On page 17, line 8, it states that "TS2 was altered to use RCIM assumptions for formation and loss of PAN and: : :" What specifically was altered in this sensitivity test?**

We remove "which are explicitly listed in Table S7" and add in a more complete sentence stating: "The revised reactions for each of these sensitivity tests are listed explicitly in Table S7." And added:
"Unlike TS2, RCIM does not include PAN photolysis or the $CH_3CO_3 + CH_3CO_3$ reaction and RCIM uses different reaction rate constants than TS2 for PAN formation, thermal decomposition, and reaction with OH (Table S7)."

**8. On page 19, line 18, it states "Commonly, in MCM, unsaturated hydroxy nitrates will react with OH via H-abstraction : : :". Please add "derived from terpenes" after nitrates to specify that this does not involve hydroxy nitrates derived from isoprene.**

Very good clarification, thanks this has been added in this location and a couple others for clarity.

**9. Figure 9 should have same color scheme for TS1 and TS2 with its predecessors (Figs 3-6) for consistency.**

This has been updated TS1 is red and TS2 is purple in all figures now.

**10. The equation used to derive the organic nitrate yields using alpha and n in the Supporting Information (pgs. 49-51) should be stated with the reference included for clarity.**

We updated the description in the Table S6 to "see notes [a]" with a link, so that it is clear that more information is in the table notes. Then we also update the note to include the formula used along with the reference.

**Response to Review #2**

Thank you for the helpful comments and suggestions. We appreciate your time for reviewing our paper. We have addressed all of your comments as detailed below:

**Overall**

**The mechanism is expected to be included in future versions of the CESM model and is offered to any interested party if they'd like to access the updates prior to the new release. If for whatever reason the TS2 mechanism does not make it into a future CESM release the authors should guarantee an alternative location for the code to be made available.**

We have updated the code data availability section to include more details. The TS2 chemistry will very likely be released as a new compset in CESM 2.2. We include reference to the CAM-chem wiki, which will always have the updated status of the new and old chemical mechanisms available in CAM-chem. If for some very unlikely reason, TS2 is not implemented into CESM 2.2, we will provide directions for how to implement the changes on this referenced wiki-page. We do not want to upload the code now as the updates include both a new chemical mechanism and source code modifications, and we do not want users to inaccurately implement the code. We also upload ozone hourly and SEAC$^4$Rs flight track data used in this work to a repository and describe this here:

"The code updates described in this work will be made available as a new compset in a future release of CESM likely CESM 2.2. Chemical mechanisms used in CAM-chem are listed on the CAM-chem wiki page at https://wiki.ucar.edu/display/camchem/Gas-Phase+Chemistry, which includes a brief description, a citation reference, and information on their current status. Please contact Rebecca Schwantes (rschwant@ucar.edu) if you would like the TS2 updates prior to their future release in CESM. CESM/CAM-chem output for 2013 August hourly ozone, 2013 August monthly average default output, and SEAC$^4$RS flight tracks used in this work are provided on NCAR's Digital Asset Services Hub (DASH) here: https://dashrepo.ucar.edu/dataset/68_rschwant.html (Schwantes and Emmons 2020)."

**Comments**

**Figures 1 and 2 should point to Table S2.**

The following has been added to the caption for Figure 1 and 2:
"All species names used in TS2 are described in Table S2."

**Figures 1 and 2 are schematics of OH oxidation and Figures S1 and S2 are schematics of NO3 oxidation. Why were no schematics of O3 oxidation included (and only described in Sections 2.2.2 and 2.3.2). For completeness, I feel that O3 oxidation schematics could be included in the supplement.**

For clarity we have added a schematic for O$_3$ oxidation of isoprene and terpenes to the Supplement in Figure S3. We add reference to Figure S3 in Section 2, 2.2.2, 2.3.2.

"and for O$_3$-initiated oxidation in Figures S3."
"O$_3$-initiated oxidation of isoprene is simply described in Figure S3"
"O$_3$-initiated oxidation of the terpene surrogate compounds in TS2 is described in a simplified schematic in Figure S3."

**Page 7, Line 1: The authors mention that "multiple isomers were only incorporated into TS2 if grouping them together would bias HOx or NOx budgets." How was this determined? Were there sensitivity or testing runs conducted and that are not described? Please describe.**

We have reworded this and added more detail. We did not run sensitivity tests for this. Instead this was determined based on the chemistry.
"To save computational cost, multiple isomers were only incorporated into TS2 if grouping them together would bias the HO$_x$ or NO$_x$ budgets. For example, the two dominant isomers of ISOPOOH react with OH to produce similar yields of OH and a similar distribution of IEPOX isomers (Wennberg et al. 2018), so grouping isomers of ISOPOOH and IEPOX together to reduce computational cost is warranted. In contrast, TS2 includes multiple isomers of HPALDs and hydroxy nitrates because different isomers react with OH to produce varying levels of OH and NO$_2$, so combining these isomers together would inaccurately influence the HO$_x$ and NO$_x$ budgets."

**Page 7, Line 9: The authors mention that the "number of organic nitrates in TS2 was optimized to accurately represent" varying reaction rates. What sort of optimization was done? What were the optimization criteria? This needs more detail.**

Good point, we updated this sentence below. Specifically, we changed "optimized" to "selected" here because there was not an optimization program run. The selection was instead based on reaction rates, products, and atmospheric fate of each isomer.
"The organic nitrates in TS2 were carefully selected to account for these varying fates, but also when possible to combine isomers with similar reaction rates, oxidation products, and overall atmospheric fate together to reduce computational cost."

**The supplemental Figures S1 and S2 should be referenced more thoroughly in the manuscript, esp. within the captions of Figures 1 and 2 and in the accompanying sections (Section 2.2.3 and 2.3.3).**

We add additional references to Figures S1 and S2 in Section 2.2.3 and 2.3.3.
"Isoprene NO$_3$-initiated oxidation in TS2 is described in a simplified schematic in Figure S1 and …"
"Where structurally similar, organic nitrates from OH and NO$_3$ oxidation are shared to reduce computational cost (Figure 1 and S1).
"as summarized in a simplified schematic in Figure S2."
"… and TERPNPS1 (unsaturated primary/secondary) (Figure S2)"
"… the corresponding saturated nitrooxy hydroperoxide (Figure S2)"

As suggested, we also now reference the $NO_3$ and $O_3$ supplemental figures in the caption of Figure 1 and 2 to make certain the reader knows these figures are available.

"Similar schematics for $NO_3$- and $O_3$-initiated oxidation of isoprene are provided in Figure S1 and S3 in the supplement."

"Similar schematics for $NO_3$- and $O_3$-initiated oxidation of terpenes are provided in Figure S2 and S3 in the supplement."

**Page 10, Line 10: Were there any alternative approaches that could be made to represent terpene products here? Were these approaches tested or considered? A description of why this particular choice was made (and what the likely impact on computational cost and resulting chemistry would be) should be included.**

We did not test many different versions of the TS2 mechanisms, but agree that in the future creating reduced chemical mechanisms more systematically from explicit chemical mechanisms would be useful. We add the following to describe in more detail why the oxidation products selected here were used.

"The terpene products described above were selected, so that compounds that are chemically similar (i.e. contain the same functional groups and react with OH, $O_3$, and $NO_3$ at similar rates) are grouped together. The complexity of the chemistry is largely determined by the current knowledge of the system. There may be advantages to adding more complexity into TS2 in the future as more knowledge about terpene oxidation especially later-generation chemistry becomes available. Given terpene-later-generation chemistry is not well understood and that chemistry of many terpene products is largely estimated based on similar more well-studied compounds, adding separate products for each of the terpene surrogate compounds would increase cost without adding a lot of additional information into the system."

**Section 2.4: I feel that a more thorough descriptions of the impacts of the TS2 mechanism on aerosol formation can be included. The existing section is sparse compared to the other sections, and although the authors make not of the large uncertainties that remain, I feel that more can be added. Perhaps the manuscript title can include the word "gas-phase" after the word "Comprehensive" in order to highlight that this manuscript is largely focused on the gas-phase chemistry of isoprene and terpene and that future development of the aerosol scheme is slated for future work.**

Yes, good point we add "gas-phase" to the title. This work is more focused on the gas-phase chemistry. The Volatility Basis Set scheme that forms SOA in CAM-chem was not changed in this work. We update Section 2.4:

"In TS2, aerosol uptake of organic nitrates is only a gas-phase sink for organic nitrates and does not form SOA directly. SOA in TS2 only forms from a VBS scheme (Tilmes et al., 2020), which was not updated in this work. Better connecting gas-phase chemistry and SOA formation is a goal for future work as uptake of organic nitrates to form SOA is important for accurately representing gas-phase ozone as well as the overall magnitude of SOA. Aerosol uptake of organic nitrates is quite uncertain (Section 5.2) and further studies investigating the processes of

organic nitrate uptake and hydrolysis as well as more complex implementation of these processes in models is warranted and a goal for future work."

**The results and discussion section (Section 4) is very long (21 pages) and I feel can be split into separate sections. Perhaps the evaluation of the explicit schemes (Sections 4.1 and 4.2) and the global modeling evaluation against surface observations and field campaign data (Sections 4.3 and 4.4) can remain in Section 4, while the discussion of the organic nitrate fate (Section 4.5) and the overall discussion of uncertainties (Section 4.6) can be moved to a Discussion Section (Section 5). The existing Conclusion section (Section 5) can then me moved to Section 6.**

We have split the results and discussion section and agree this adds clarity. Thanks for the suggestion.

**In Figures 3, 4, 5, and 6, I find it somewhat difficult to differentiate between the blue colors used to label TS1 and TS2. Changing one of them to a more distinct color (green, perhaps?) would aid in clarity. [NOTE: This appears to be the case only when I print the manuscript out, and the colors are clearly distinguishable when I look at a digital copy]**

The colors have been updated such that blue and cyan are no longer included together in any of the plots, so we have updated Figures 3, 4, 5, 6, S4, S5, 9, 10, S6, S7. We have also increased the width of the lines and all Test cases for the box-model analysis are in dashed lines to make it easier to differentiate the different lines on the plot.

**The sensitivity tests in the Figures are labelled as "Test 1, Test 2", while in the captions are labelled "test 1, test 2" and in the manuscript as "first sensitivity test, second sensitivity test." I suggest sticking with the "Test 1, Test 2" format and making this consistent throughout the sections.**

Yes, we have adjusted this to refer to the sensitivity tests as a whole as "sensitivity tests" and individually as Test 1, Test 2, etc.

**Page 19, Line 26: The authors state that "These unrealistically fast cycles are commonly used in MCM to reduce the number of species...". Is there a citation for this claim? How do the authors know this is the case?**

We have seen this for other systems too (e.g., cresol oxidation pathway - Schwantes et al. 2017, ACP), but because we have not done a thorough examination and it is possible these are the only 2 instances, we remove:
"are commonly used in MCM to reduce the number of species and"

**Page 31 Line 7-8: Could you expand (speculate) on some of the possible sources of uncertainties that are less-well known?**

Yes, we have added an example:

"For example, our understanding of reaction rates and products from peroxy radical isomerization reactions that lead to auto-oxidation is rudimentary and under-constrained (Wennberg et al. 2018). Because much is unknown about peroxy radical isomerization reactions, there is not a clear understanding of how important these reactions are in the ambient atmosphere."

**Page 34, Line 2 – Page 35, Line 1: This point has also been made by Mao et al. (2018) (www.atmos-chem-phys.net/18/2615/2018/) (see page 2622), and should be referenced.**

This reference has been added.

**Page 37, Line 28: I suggest including the Sun et al. (2017) reference in which they were able to reduce Eastern US ozone bias via the utilization of a new solver scheme (https://agupubs.onlinelibrary.wiley.com/doi/full/10.1002/2016MS000863). Additionally, I believe a slightly expanded discussion of the remaining sources of uncertainty would significantly enhance this final paragraph.**

Great point. This reference has been added to the introduction and chemical solver is added as an additional uncertainty in the conclusions. Additional discussion of the remaining sources of uncertainty also has been added to the conclusion:

"This bias could be caused by remaining uncertainties in the chemistry (Section 5.2) or by processes other than chemistry, which will be evaluated in future work. Considering the analysis against the SEAC$^4$RS field campaign results, the first step toward reducing the remaining ozone bias will be to evaluate how finer horizontal resolution (14 km) impacts the results. Because biogenic and anthropogenic emissions in the southeast U.S. are spatially segregated, improvements in simulated ozone and biogenic VOCs are expected with finer horizontal resolution. Future work will also include evaluating different anthropogenic emission inventories and a more thorough investigation into whether biogenic emissions are accurately represented by MEGAN in CAM-chem. Additionally, cloud biases in CAM-chem will be investigated more in the future given their likelihood for improving the vertical profile shape of ozone, ozone precursors, and $j_{NO2}$. Considering that biases in the ozone profile shape are enhanced with stronger nudging to meteorological data (Figure S6), a more thorough analysis on the impact of nudging on CAM-chem dynamics and cloud parameterizations should be conducted. Future work will also evaluate whether enhanced vertical resolution is needed to improve PBL height and mixing schemes. Further evaluation of different chemical solvers (Sun et al., 2017) is also needed. Additionally, ozone dry deposition has a large impact on simulated surface ozone (val Martin et al., 2014; Clifton et al., 2019) and a thorough evaluation and update to the ozone dry deposition scheme used in CAM-chem should be performed. Ozone is a complicated pollutant to accurately simulate in models. This work demonstrates that updating isoprene and terpene gas-phase chemistry clearly improves simulated surface ozone in CAM-chem and that additional studies evaluating and updating other processes are needed to further reduce the ozone bias."

**Response to Review #3**

Thank you for the helpful comments and suggestions. We appreciate your time for reviewing our paper. We have addressed all of your comments as detailed below:

**1) The box model simulations, comparing TS2 with TS1 and explicit mechanisms, currently only uses one set of chemical and meteorological conditions (August in Mississippi). It would have been helpful for box model simulations to have been conducted separately for high-NOx and low-NOx conditions. Given the very different oxidation pathways under these conditions, this would have provided a more stringent set of tests for the new mechanism. (I realize that new simulations would impose a significant analysis burden on the authors, so may not be practical.)**

We do agree this would be useful. A more thorough box-modeling study comparing MOZART-TS2 against additional reduced and explicit mechanisms at varying $NO_x$ levels, VOC levels, temperatures, aerosol concentrations, photolysis levels, etc. would be extremely useful and a goal for a future study.

**2) It might be helpful to split Results (4.1-4.4) and Discussions (4.5-4.6) into separate sections.**

Great point, we have done this.

**Specific comments**

**1. Introduction**
**page 3, lines 11-12 – Rephrase for clarity. For instance, "to determine the extent to which [improvements to] the chemical mechanism can explain ...."**

Yes we update to the following:
"will be updated to determine the extent to which improvements to the gas-phase chemical mechanism for biogenic VOCs can explain the simulated surface ozone bias over the southeastern U.S."

**2.1 Updates to Henry's Law Constants**
**p.5, l.15-16 – Clarify the definition of Henry's law temperature dependence (given here as 6014 K).**

As suggested by reviewer 1 we add:
"If the Henry's law temperature dependence **(dH/R)** was unavailable in the literature, 6014 K was assumed consistent with GECKO-A."

**p.5, l. 16 – Add "used for dry deposition" after "reactivity factor (F0)."**

As suggested we update to:

"The reactivity factor ($F_0$) **used for dry deposition and ranging** from 0 to 1 with 1 being as reactive as ozone is also listed in Table S4."

**3.1 Box modeling**
**p.13, l.24 – Planetary boundary layer \*height\*?**

Yes, thanks we have added "height" here.

**p.13, l.24 – Clarify here that only "general" photolysis rate constants are taken from CESM/CAM-chem-TS1 (as explained later).**

Yes we add "general" here.

**p.13, l. 27-28 – Explain how deposition from CESM/CAM-chem is implemented in box model. For instance, dry deposition velocities? wet deposition loss frequencies? No ventilation/dilution of the box with background air is included, correct?**

Yes, thanks we have revised this to be more descriptive.
"Aerosol uptake of the following inorganic compounds: $HO_2$, $N_2O_5$, $NO_2$, $NO_3$ were included based on the reaction rate constants output from the CESM/CAM-chem base simulation. Dry deposition of the following inorganic compounds: $O_3$, CO, NO, $NO_2$, $HNO_3$, $N_2O_5$, $HO_2NO_2$, $H_2O_2$, and $SO_2$ were included using the dry deposition velocities from the CESM/CAM-chem base simulation. The box is mixed based on the planetary boundary layer height with background air, which has fixed concentrations of isoprene, terpenes, $H_2O$, $CH_4$, $H_2$, CO, $O_3$, NO, $NO_2$, $SO_2$, and $N_2O$ from the CESM/CAM-chem TS1 base simulation."

**p.14, l.13-14 – "These are ideal scenarios designed" to "These idealized scenarios are designed ...."**

Yes, we updated this.

**3.2 Global modeling**
**p.14, l.18-21 – Which meteorological fields are nudged to reanalysis?**

We update this to a more complete list:
"The meteorology (air and surface temperature, horizontal winds, surface pressure, sensible and latent heat flux, and wind stress)"

**p.15, l.4-5 – Which years were used for spinup?**

We added the following for clarity:
"separately spun-up for 2.5 years **(i.e., Jan 1, 2011 to July 31, 2013)**"

**4. Results and Discussions**
**Figures 3-6 – Difficult to distinguish some of the individual lines in these plots. Try to modify colors, or make lines thicker.**

Yes, the colors have been updated, the line width increased, and the sensitivity tests are now in dashed lines for easier viewing.

**p.17, l.8-9 – Explain the differences between the representations of PAN formation and loss (TS2 versus RCIM).**

We added the following:
"Unlike TS2, RCIM does not include PAN photolysis or the $CH_3CO_3 + CH_3CO_3$ reaction and RCIM uses different reaction rate constants than TS2 for PAN formation, thermal decomposition, and reaction with OH (Table S7)."

**p.17, l.11 – Add "from RCIM" after "The PAN assumptions."**

Added "from RCIM"

**p.17, l.13 – Are the RCIM photolysis rates faster or slower? By how much?**

This is explained in the next paragraph. We rearrange these paragraphs, so that this is clearer.

**4.2 Terpene Evaluation Against Explicit Species**
**p.19, l.5 – Clarify what is meant here by "total products produced."**

We revise this sentence to the following:
In general, the types of compounds formed are their concentrations are reasonably consistent between MCM and TS2.

**p.20, l.3 – Add "oxidation of" before "the alpha-pinene."**

Yes, added this.

**4.4 Evaluation Against Field Campaign Data**
**Figure 7 – Add mean bias for Eastern US / Western US to figure panels.**

The mean bias has been added here.

**p.28, l.1-3 – How do the dry deposition velocities of OVOCs compare in GEOS-Chem versus CESM/CAM-chem?**

I do not have data for the dry deposition velocities of OVOCs in the standard version of GEOS-Chem to compare to my data with CAM-chem, so I cannot make this comparison at this time. However, more comparisons between GEOS-Chem and CAM-chem in the future would be really useful and hopefully can be performed in the near future. We are in the process of evaluating a new version of CESM/CAM-chem with finer horizontal resolution down to 14 km. This future study will include a comparison for OVOC dry deposition velocities measured during SOAS with CAM-chem results.

**4.6.2 Uncertainties in Loss of Organic Nitrates**
**p.35, l.24 and l.26 – Clarify the meaning of "largely" here, e.g., do you mean "primary and secondary organic nitrates *largely* will not" and "... are *largely* lost ...."**

Yes, good point this "largely" should be next to the verb for clarity. I move this and change the first instance to "generally".

**Technical corrections**
**1. Introduction**
**p.3, l.3 – Hyphenate "terpene-derived" (and "isoprene-derived" throughout manuscript).**

Yes, we updated this.

**2. Development of MOZART-TS2**
**p.3, l.11 – Capitalize "Model."**

Yes, updated.

**3.1 Box modeling**
**p.14, l.6 – Include units for lat/lon (deg N, deg E).**

Yes, added these units.

**3.2 Global modeling**
**p.14, l.22 – Run-on sentence. Break into two, starting with "Using a weak ...."**

Yes, this sentence was revised:
"This study uses 32 vertical levels and a weak relaxation time (50 h) for nudging in order to reduce variability while also limiting the impact of nudging on model parameterizations."

**p.14, l.34 – Change to "(Table S3), using ...."**

We realized that the description for vertical levels may not be clear enough. We have added more detail here and in the paragraph above:

"First, the ``TS1" case uses the default CESM2.1.0/CAM-chem code and the default TS1 chemical mechanism **with the two changes described above: 32 vertical levels and** an expansion of the biogenic volatile organic compounds emitted from the land model (Table S3)"

And in the paragraph above:
"Using 32 vertical levels, the vertical resolution to which CAM physics, dynamics, and cloud parameterizations are tuned, slightly improves the model bias for ozone near the surface compared to using 56 vertical levels, the native resolution of the MERRA2 meteorological files (Figure S6)."

**4. Results and Discussions**
**p.15, l.15 – "suggests" –> "suggest"**

Yes, this is fixed.

**p.17, l.1 – Hyphenate "NO3-initiated."**

Yes, this is fixed.

**4.2 Terpene Evaluation Against Explicit Species**
**p.20, l.1 – "Terpene-rich"**

Yes, this is fixed.

**4.4 Evaluation Against Field Campaign Data**
**p.27, l.3 – Change to "above 2km; when clouds ...."**

Yes, this is fixed.

**4.5 Organic Nitrate Formation and Fate**
**p.28, l.34 – "isoprene- and terpene-derived."**

Yes this is fixed throughout.

**p.29, l.6 – Delete comma.**

Yes, this is fixed.

**4.6.1 Uncertainties in Formation of Organic Nitrates**
**p.33, l.6 – Delete comma.**

Yes, this is fixed.

**4.6.2 Uncertainties in Loss of Organic Nitrates**
**p.35, l.13 – Delete comma after "(Figure 1)."**

I removed comma and put description in parenthesis instead to avoid confusion.

**p.35, l.16 – "under-constrained, leading to ...."**

Yes, this is fixed

**5. Conclusions**
**p.36, lines 15, 18, 31 – Missing commas.**

Yes, this is fixed

**p.37, line 9 – Missing comma.**

Yes, this is fixed

[revised manuscript text omitted]

- [a] For all isoprene peroxy radicals, the $RO_2$ + NO general reaction rate constant (2.7e-12 exp(360/T)) from MCM v3.3.1 is used with the pressure and temperature dependent organic nitrate yield recommended by Wennberg et al. (2018), which is based on Arey et al. (2001). The organic nitrate yield ($\alpha$) at T = 293 K and M = 2.45e19 molecules $cm^{-3}$ and the number of heavy atoms excluding the peroxy group (n) are reported above. $\alpha$ and n are used in the following equation to calculate the temperature and pressure dependent organic nitrate branching ratio ($\alpha\_RONO2$):

$$\alpha\_RONO2 = \frac{A(T,M,n)}{A(T,M,n)+A_0(n)*\frac{1-\alpha(n)}{\alpha(n)}}, \text{ where } A_0(n) = A(T=293K, M=2.45e19, n),$$

$\gamma$ = 2e-22 $cm^3$ $molecule^{-1}$, T = temperature (K), [M] is the number density of air in molecules $cm^{-3}$,

and $A(T,M,n) = \dfrac{\gamma e^n[M]}{1+\frac{\gamma e^n[M]}{0.43(T(K)/298)^{-8}}} * 0.41^{(1+[log(\gamma e^n[M]/0.43(T(K)/298)^{-8})]^2)^{-1}}$

[revised manuscript text omitted]

---

## Author Response (AR2)

As suggested by the editor, minor changes to the text were made to include an additional 2 references. We also updated a couple typos. None of these changes significantly alter the content of the paper.

The latexdiff between the previous uploaded version (Jan 24[th] 2020) and this new uploaded version (Feb 14, 2020) are provided below.

[revised manuscript text omitted]